# Analytical and adaptable initial conditions for dry and moist baroclinic waves in the global hydrostatic model OpenIFS (CY43R3)

Clément Bouvier[1], Daan van den Broek[1], Madeleine Ekblom[1], and Victoria A. Sinclair[1]

[1]Institute for Atmospheric and Earth System Research/Physics, Faculty of Science, University of Helsinki, Finland

**Abstract.** This article presents a description of an analytical, stable and flexible initial background state for both dry and moist baroclinic wave simulation on an aquaplanet in order to test dynamical core of numerical weather prediction models and study the dynamics and evolution of extra-tropical cyclones. The initial background state is derived from an analytical zonal wind speed field, or jet structure, and the hydrostatic primitive equations for moist adiabatic and frictionless flow in spherical coordinates. A baroclinic wave can develop if a perturbation is added to the zonal wind speed field. This new baroclinic wave configuration has been implemented in the Open Integrated Forecasting System (OpenIFS) CY43R3, a global numerical weather prediction model developed by the European Centre for Medium-range Weather Forecasts. In total, seven parameters can be used to control the generation of the initial background state and hence the development of the baroclinic waves in the OpenIFS configuration file: the jet's width, the jet's height, the maximum zonal mean wind speed of the jet, the horizontal mean of the surface virtual temperature, the surface relative humidity, the lapse rate and the surface roughness. Nine dry and nine moist initial background states have been generated to test their stability without perturbations. The meteorological stability of the initial states are investigated by examining the spatial distributions of the equivalent potential temperature, the absolute vorticity and the Brunt-Väisälä frequency. Moreover, the Root-Mean-Squared-Error (RMSE) of the zonal wind speed has been computed to assess their numerical stability. Finally, six dry and six moist initial background state have been used with an unbalanced perturbation to ensure that the baroclinic lifecycles that develop are physically realistic. The resulting baroclinic wave is shown to be sensitive to the jet's width. This configuration for baroclinic wave simulations will be used to create a large ensemble of baroclinic lifecycles to study how extra-tropical cyclones may evolve in the future.

## 1 Introduction

General-Circulation Models (GCMs) are an important tool to predict the extent of global climate change as documented in the IPCC reports (IPCC, 2022). These GCMs provide numerical solutions to the governing equations of the atmosphere. They can take into consideration real-world data to predict the short term evolution of the weather or they can be used to simulate idealised weather systems to study specific phenomena of our climate such as convection (Khairoutdinov et al., 2022) or baroclinic waves (Ullrich et al., 2015). Baroclinic waves are the synoptic-scale patterns of high and low pressure systems that develop in the mid-latitudes. These waves develop due to the release of baroclinic instability and the resulting patterns are important parts of the Earth's global circulation as they transport energy polewards (Simmons and Hoskins, 1978; Thorncroft et al., 1993; Beare, 2007).

Two main reasons exist to perform Baroclinic Wave Simulation (BWS). To further improve our weather prediction and climate models, the dynamical cores have to be tested. Most BWS experiments specify a zonally uniform solution to the hydrostatic primitive equations that is statically stable and stable to inertial and symmetric instabilities and then run a numerical model with this specified as the initial state. This type of simulation tests the ability of the numerical model to retain this exact solution in the presence of numerical errors. These initial states are baroclinically and barotropically unstable and therefore adding a perturbation triggers the development of a baroclinic wave - to which there is no exact solution (Hoskins and Simmons, 1975; Simmons and Hoskins, 1975; Jablonowski and Williamson, 2006). The baroclinic wave development can be simulated on an f-plane or, its extension, a $\beta$-plane in both Cartesian and spherical geometries (Feldstein and Held, 1989; Staniforth and White, 2007; Ullrich et al., 2015). These models are often less expensive to run from a computational point of view by simplifying the Coriolis forces. Another approximation that is used alongside f- and $\beta$-planes is the restriction of the size of the model domain where the zonal extent of the domain is set roughly equal to the most unstable wavelength (~4000 km) (Hoskins et al., 1977; Ullrich et al., 2015). With this limitation, any upstream or downstream development is forced to occur on top of the main perturbation. This representation can efficiently display the energy propagation of the baroclinic wave (Hoskins et al., 1977). However, it becomes increasingly difficult to study dynamical and synoptic properties of the cyclones without the realistic simulation of upstream and downstream developments. Moreover, numerous description and specification of the initial states are available for Cartesian geometry and for channel models (Hoskins et al., 1977; Feldstein and Held, 1989; Wang and Polvani, 2011; Ullrich et al., 2015; Terpstra and Spengler, 2015) and for spherical geometry and fully global models (Polvani et al., 2004; Jablonowski and Williamson, 2006; Staniforth and White, 2011; Ullrich et al., 2014; Hughes and Jablonowski, 2023), but none proposes an initial state with tunable parameters (e.g., defining the width of the jet), which is one of the main motivations for this work.

The Baroclinic Wave Simulations are of interest to study extra-tropical cyclones, extreme cases which will likely become more frequent in the future (IPCC, 2022). Extreme cyclones are characterised by strong winds, heavy precipitation and powerful ocean waves. Consequently, these extreme events can damage infrastructure, forests, homes, cause flooding and result in injuries and even death. Depending on the location and the state of the large-scale background environment that the cyclone develops in, the structure and intensity of a given cyclone can vary considerably (Tang et al., 2020). Traditionally, the BWS were adiabatic and the simulations were run without physics parameterisation schemes (Simmons and Hoskins, 1978; Thorncroft et al., 1993). The main reason being that the synoptic-scale dynamics of baroclinic waves can be largely explained by the classic quasigeostrophic theories of dry baroclinic instability (Charney, 1947; Eady, 1949). Moreover, the impact of latent heat on extra-tropical cyclone intensity has been heavily investigated in the last three decades (e.g., Kuo et al., 1991; Stoelinga, 1996; Willison et al., 2013; Park et al., 2021). For example, diabatic processes are important to the evolution of the precipitation (Kuo et al., 1991; Park et al., 2021) and smaller-scale systems (Stoelinga, 1996). Moreover, to predict how cyclones may change in the future, it is necessary to determine how sensitive baroclinic waves are to changes in temperature and moisture. This motivated the development of new BWS which were designed to be run with physics parameterisation schemes acting and moisture present (Beare, 2007; Kirshbaum et al., 2018; Tierney et al., 2018; Rantanen et al., 2019).

The sensitivity of the resultant extra-tropical cyclones to the jet structure has been studied (Thorncroft et al., 1993; Shapiro et al., 1999; Rupp and Birner, 2021). Popular zonal jet structures are Zonal jet 1, Zonal jet 2 and Zonal jet 3 (denoted Z1, Z2 and Z3) resulting in, respectively, baroclinic lifecycles 1, 2 and 3 (denoted LC1, LC2 and LC3) (Thorncroft et al., 1993; Agustí-Panareda et al., 2005). Z2 and Z3 differs from the zonally quasi-symmetric jet of Z1 by including a cyclonic (Z2) or anti-cyclonic (Z3) barotropic shear (Thorncroft and Hoskins, 1990; Thorncroft et al., 1993; Shapiro et al., 1999; Polvani and Esler, 2007). Depending on the barotropic shear, baroclinic lifcycles have different structures and intensities (Agustí-Panareda et al., 2005). However, it is difficult to control the height and the width of the jet and test the sensitivity of the BWS to these parameters due to the finite amount of jet structures tested. Here, an initial state is developed, in which the jet's width and vertical structure can be varied in addition to the jet strength. However, setting up a balanced and flexible background state with several jet structures is difficult. The challenges lie in balancing the initial conditions at high resolutions in state-of-the-art models. Few cases are fully documented and can be difficult to reproduce. Many are based on the model developed by Hoskins and Simmons (1975) and rely on numerical integration which is prone to truncation errors. Some are based on Cartesian geometry as presented in Kirshbaum et al. (2018) and their jet structures are obtained from the potential vorticity inversion method (Heckley and Hoskins, 1982; Olson and Colle, 2007). Having an analytical structure of the jet may allow more control on its structure and strength.

The aim of this study is to describe a balanced, flexible, initial background state for a baroclinic life cycle experiment that can be entirely expressed analytically and that produces relatively realistic weather systems. The analytical solution is derived from a steady-state momentum equation for the meridional wind speed. In other words, the meridional wind speed is set to $0.0 \text{ ms}^{-1}$ which leads to a gradient-wind balance. Moreover, the proposed background state is also in hydrostatic balance, i.e., the hydrostatic equation was used to derive the virtual temperature anomaly from the geopotential field. The initial background state is also based on a flexibly defined jet structure and can furthermore be initialised with moisture by changing the relative humidity profile. The theoretical description and derivation of the initial state with mathematical formulae together with the method for including moisture is presented in section 2. The technical implementation into the global, state-of-the-art numerical weather prediction model, the Open Integrated Forecasting System (OpenIFS), is described in section 3. The different experiments are described in section 4 and their associated results in section 5. First, the new initial states (both dry with no physics parameterisation schemes and moist with almost all physics parameterisation schemes) are run for 15 days with no perturbation to confirm that the initial states are indeed stable both numerically and from a meteorological perspective. Second, six dry and six moist initial background states have been used with an unbalanced perturbation to generate baroclinic life cycles. The evolution of the resulting baroclinic waves that develop from our default dry and moist initial state are shown. The results from the moist, default simulation also show how the precipitation patterns form as the cyclones develop.

## 2   Initial Condition for the Baroclinic Wave

This section presents a balanced, steady-state, initial condition for a 3D hydrostatic atmospheric model in spherical coordinates with a flexible jet structure. The moisture field is defined to be consistent with the virtual temperature field. This section

describes the theoretical background for the initial conditions and is divided into four parts: (1) analytical derivation of the geopotential and virtual temperature fields, (2) initialisation of moisture, (3) initialisation of the surface (both sea-surface temperature (SST) and roughness), and (4) description of the unbalanced perturbation, which when added triggers the development of the baroclinic wave.

## 2.1 Analytical Geopotential and Virtual Temperature Fields

The derivation of the analytical initial conditions for geopotential and virtual temperature fields starts from the primitive equations for moist adiabatic and frictionless flow in spherical coordinates and normalised pressure levels for a planet with no topography (i.e., surface geopotential is zero). The geopotential and virtual temperature anomaly fields are derived from hydrostatic equations, and the derived initial states apply to both hydrostatic and non-hydrostatic models. The geopotential and virtual temperature fields are described as the horizontal mean field as a function of vertical levels plus an anomaly field which is a function of longitude, latitude and vertical levels (respectively $\lambda, \phi, \eta$). In OpenIFS, the vertical $\eta$ levels are defined as $\eta = p/p_s = a/p_s + b$, where $p_s = 1013.25\text{hPa}$ is the pressure at the surface pressure, and $a$ and $b$ are hybrid coefficients defined for each vertical resolution. The horizontal means proposed by Ullrich et al. (2015) are used in this background state. The analytical formula of the horizontal mean geopotential is described as

$$\langle \Phi(\eta) \rangle = \frac{T_{v,0}\, g}{\gamma}(1 - \eta^{\frac{R_d \gamma}{g}}) \tag{1}$$

and the horizontal mean virtual temperature field as

$$\langle T_v(\eta) \rangle = T_{v,0}\, \eta^{\frac{R_d \gamma}{g}}, \tag{2}$$

where $\gamma$ is the specified lapse rate, $T_{v,0}$ the reference virtual temperature, $R_d$ the gas constant for dry air and $g$ the gravity constant. The derivation for the horizontal mean geopotential is available in the Appendix.

To be able to derive the anomaly fields of geopotential and virtual temperature, a jet structure has been defined similar to the one proposed by Ullrich and Jablonowski (2012) and Ullrich et al. (2015) with the only difference being the power of the sine function. The power of the sine is described as $2n$ allowing for a narrower jet when $n$ increases. The chosen formula can be expressed as

$$u(\lambda, \phi, \eta) = -u_0 \ln(\eta) \exp[-(\frac{\ln \eta}{b})^2] \sin^{2n}(2\phi), \tag{3}$$

where $n$ is a positive integer defining the width of the jet, $b$ is a non-dimensional parameter representing the depth of the jet, and $u_0$ is the reference zonal wind speed and defines the zonal-mean speed of the jet in the troposphere. As expressed by Eq. (3), the jet width decreases with an increase of $n$, and the height of the centre of the jet and the vertical width of the jet increase with increasing values of $b$. The jet reaches its maximum wind speed at $\phi = 45°$ and $\eta = \exp(-b/\sqrt{2})$. Furthermore, the value for $b$ needs to be positive and smaller than $-\sqrt{2}\ln(\eta_{top})$, where $\eta_{top}$ is the ratio between the top pressure level and the surface pressure level. For example, a top pressure level of $0.01$ hPa and surface pressure level of $1000$ hPa, the upper limit for $b$ is about 16. If the value of $b$ exceeds the upper limit, the centre of the jet is located outside the model domain. The analytical

geopotential and virtual temperature fields are solved for any jet structure defined by Eq. (3), i.e., for arbitrary values of $n$ and $b$.

The derivation of the geopotential and virtual temperature anomaly fields start from the primitive equations for moist adiabatic and frictionless flow (Holton and Hakim, 2013). Following the instructions given in Appendix A of Jablonowski and Williamson (2006), the geopotential anomaly field has been derived from the steady-state momentum equation for the merid-

ional flow ($\partial v/\partial t = 0$) by inserting our choice of jet structure and solving for $\Phi'(\lambda, \phi, \eta)$

$$\frac{1}{a}\frac{\partial \Phi'}{\partial \phi} = -u\left(2\Omega\sin\phi + \frac{u}{a}\tan\phi\right), \tag{4}$$

where $\Omega$ is the angular velocity of the Earth, $u$ the jet structure given by Eq. (3), and $a$ the radius of the Earth. A steady-state solution leads to a gradient wind balance, where the centrifugal, Coriolis and pressure gradient forces are in balance. Integrating Eq. (4) analytically over $\phi$ results in

$$\Phi'(\lambda, \phi, \eta) = -u_\eta 2a\Omega 4^n \sum_{k=0}^{n}\binom{n}{k}(-1)^k \frac{1}{2(k+n)+1}\cos^{2(k+n)+1}\phi \tag{5}$$

$$- u_\eta^2 16^n \sum_{k=0}^{2n-1}\binom{2n-1}{k}(-1)^k \frac{1}{2(k+2n+1)}\sin^{2(k+2n+1)}2\phi$$

$$+ \Phi_0(\eta).$$

Since the deviations of $\Phi'$ vanishes when averaging horizontally, $\Phi_0$ is solved by inserting $\Phi'(\lambda, \phi, \eta)$ in the horizontal mean equation

$$\frac{1}{4\pi}\int_0^{2\pi}\int_{-\pi/2}^{\pi/2}\Phi'(\lambda, \phi, \eta)\cos\phi \, d\phi \, d\lambda = 0, \tag{6}$$

which then gives the analytical geopotential anomaly field $\Phi'$ as

$$\Phi'(\lambda, \phi, \eta) = u_\eta a\Omega 4^n\left(F_3 - 2F_1\right) + u_\eta^2 16^n\left(\frac{1}{2}F_4 - F_2\right), \tag{7}$$

where

$$F_1 = \sum_{k=0}^{n}\binom{n}{k}(-1)^k \frac{1}{2(k+n)+1}\cos^{2(k+n)+1}\phi, \tag{8a}$$

$$F_2 = \sum_{k=0}^{2n-1}\binom{2n-1}{k}(-1)^k \frac{1}{2(k+2n+1)}\sin^{2(k+2n+1)}2\phi, \tag{8b}$$

$$F_3 = \sum_{k=0}^{n}\binom{n}{k}(-1)^k \frac{1}{2(k+n)+1}\sqrt{\pi}\frac{\Gamma(k+n+3/2)}{\Gamma(k+n+2)}, \tag{8c}$$

$$F_4 = \sum_{k=0}^{2n-1}\binom{2n-1}{k}(-1)^k \frac{1}{2(2n+k+1)}\frac{2}{2(2n+k+1)+1} \quad \text{and} \tag{8d}$$

$$u_\eta = u_0 \ln\eta \exp(-[\ln\eta/b]^2). \tag{8e}$$

Note that $\binom{n}{k}$ is the binomial coefficient representing the $k$ unordered outcomes from $n$ possibilities, $\Gamma(x)$ is the Gamma function for positive half-integer $x = z + 1/2$ with $z$ a positive integer and $2n$ is the power of the sine in the jet structure.

The total geopotential field is described as the sum of the mean horizontal geopotential field and the anomaly geopotential field as

$$\Phi(\lambda, \phi, \eta) = \frac{T_{v,0}g}{\gamma}(1 - \eta^{\frac{R_d\gamma}{g}}) + u_\eta a\Omega 4^n \left(F_3 - 2F_1\right) + u_\eta^2 16^n \left(\frac{1}{2}F_4 - F_2\right). \tag{9}$$

The virtual temperature anomaly field is then derived by inserting $\Phi'(\lambda, \phi, \eta)$ into the hydrostatic equation and taking the derivative of $\Phi'$ with respect to $\eta$

$$T_v'(\lambda, \phi, \eta) = -\frac{\eta}{R_d}\frac{\partial \Phi'(\lambda, \phi, \eta)}{\partial \eta}, \tag{10}$$

which gives the virtual temperature field

$$\begin{aligned}
T_v(\lambda, \phi, \eta) &= \langle T_v(\eta)\rangle + T_v'(\lambda, \phi, \eta) \\
&= T_{v,0}\eta^{\frac{R_d\gamma}{g}} + \frac{u_0}{R_d}\exp[-(\ln\eta/b)^2]\left[\frac{2(\ln\eta)^2}{b^2} - 1\right]\left[a\Omega 4^n\left(F_3 - 2F_1\right) + 16^n u_\eta\left(F_4 - 2F_2\right)\right],
\end{aligned} \tag{11}$$

where $F_1, F_2, F_3, F_4$ and $u_\eta$ are as defined in Eq. (8). A detailed step-by-step derivation of the analytical geopotential and temperature anomaly fields is available in Appendix A.

## 2.2 Moisture Initialisation

As stated in the Introduction, the proposed background state can be used in dry and moist cases studies. In order to set the latter, a relative humidity profile with respect to water $RH(\eta)$, depending on the model level $\eta$ and the surface relative humidity $RH_0$, has been defined. It is inspired from the ERA-Interim (Romps, 2014) and ERA5 (Gamage et al., 2020) average relative humidity profile. The profile, as shown in Figure 1, has a maximum value of the $RH_0$ at the surface, above which it decreases to 70 % of $RH_0$ at $\eta = 0.8$. Between 0.8 and 0.3, RH is constant and for $\eta < 0.3$ it again decreases linearly to 0 at $\eta = 0.1$. Above 0.1, RH is set to 0 %. The profile is given as

$$RH(\eta) = \begin{cases} 0.0\% & \text{between } \eta = 0 \text{ and } 0.1 \\ (3.5\eta - 0.35)RH_0 & \text{between } \eta = 0.1 \text{ and } 0.3 \\ 0.7RH_0 & \text{between } \eta = 0.3 \text{ and } 0.8 \\ (1.5\eta - 0.5)RH_0 & \text{above } \eta > 0.8. \end{cases} \tag{12}$$

The specific humidity field $q(\lambda, \phi, \eta)$ is then computed to ensure concordance with the proposed virtual temperature and jet structure by assuming $T = T_v$. The specific humidity field is derived from the relative humidity ($RH(\eta)$), the saturation vapour pressure ($e_s$) and the saturation mixing ratio ($w_s$) using the Bolton approximation for the saturation vapour pressure (Bolton,

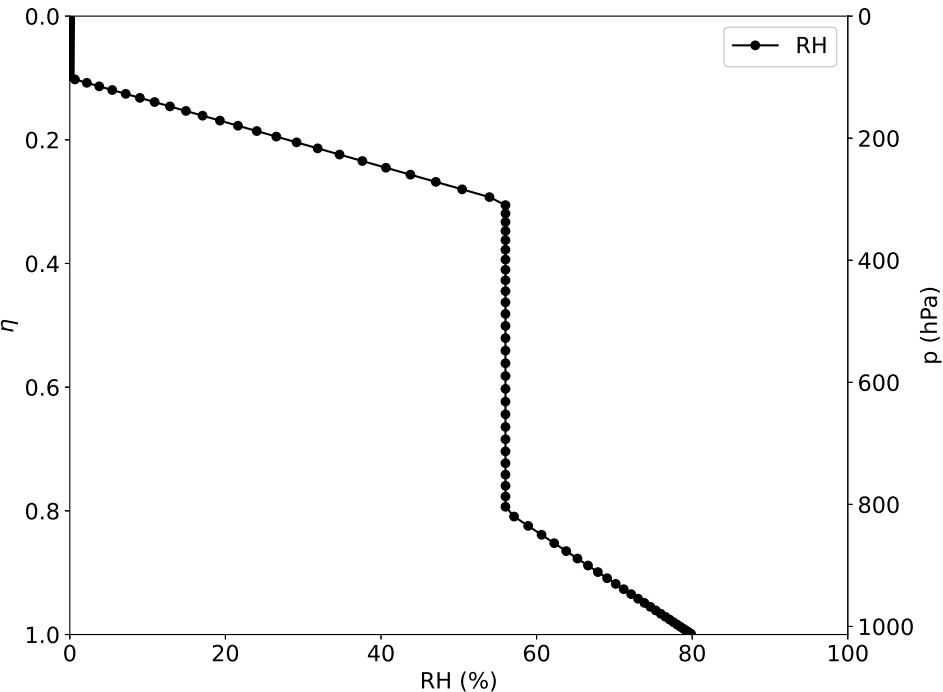

**Figure 1.** Relative humidity profile for $RH_0 = 80\%$ as a function of height. The left-hand y-axis shows $\eta$-levels and the right-hand y-axis the corresponding pressure levels in hPa.

1980; Yau and Rogers, 1996) as presented in the following equations

$$e_s(\lambda, \phi, \eta) = 611.21 \exp \frac{17.67(T(\lambda, \phi, \eta) - 273.15)}{T(\lambda, \phi, \eta) - 29.65} \text{ and} \tag{13a}$$

$$\text{175} \quad w_s(\lambda, \phi, \eta) = 0.622 \frac{e_s(\lambda, \phi, \eta)}{p(\lambda, \phi, \eta) - e_s(\lambda, \phi, \eta)}, \tag{13b}$$

where $T(\lambda, \phi, \eta)$ is the temperature field (K), $e_s(\lambda, \phi, \eta)$ is the saturation vapour pressure (Pa) and $p$ is the pressure (Pa). Finally, $w_s(\lambda, \phi, \eta)$ and $RH(\eta)$ are used to infer $q(\lambda, \phi, \eta)$ as

$$q(\lambda, \phi, \eta) = \frac{w_s(\lambda, \phi, \eta) RH(\eta)}{w_s(\lambda, \phi, \eta) RH(\eta) + 100\%}. \tag{14}$$

The formulations of the virtual temperature and the specific humidity lead to the following expression for the temperature field

$$\text{180} \quad T(\lambda, \phi, \eta) = \frac{T_v(\lambda, \phi, \eta)}{1 + 0.608q(\lambda, \phi, \eta)}. \tag{15}$$

The process for updating the temperature and the specific humidity needs to be computed iteratively. The iteration starts by first setting $T=T_v$ after which $e_s$ is computed using the Bolton equation (Eq. (13a)) and $w_s$ and $q$ are computed using equations

(13b) and (14), respectively. When $q$ is updated, then $T = T(T_v, q)$ is updated. In the following iteration, the estimated $T$ is again used to estimate a new $T$. Tests show that this iterative process converges quickly and after 10 cycles, the algorithm has reached an optimum. The final result is $T$ and $q$. Figure 2 shows how this iterative process works for computing $T$ and $q$.

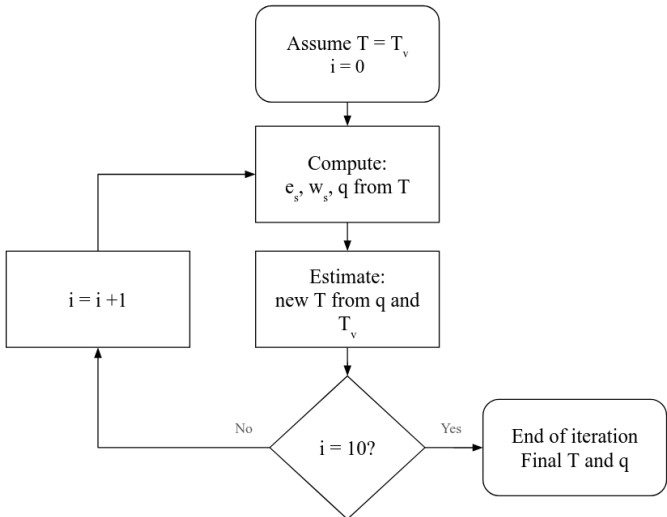

**Figure 2.** Flowchart showing how to compute $T$ and $q$. During each iteration $e_s$, $w_s$ and $q$ are computed from $T$ after which the temperature $T$ is updated using the new $q$ and the virtual temperature $T_v$. The iterative process ends when the number of iterations is 10 and the final $T$ and $q$ are returned.

## 2.3 Surface Initialisation: Temperature and Roughness

A uniform sea surface with no land has been chosen for the described background state. Thus, the experiments presented here were conducted using an aquaplanet setting which is the traditional configuration of baroclinic wave simulation (Jablonowski and Williamson, 2006; Ullrich et al., 2015). The Sea Surface Temperature (SST) is zonally uniform and is specified to equal the temperature field at $\eta = 1$ (see Eq. (11) and (15)), which means negative temperatures are allowed and the zonal wind is equal to $0.0 \text{ ms}^{-1}$. If the SST differed from the near surface atmospheric temperature, then in the moist cases there would be non-zero surface sensible heat fluxes which could either heat or cool the boundary layer. Such fluxes could trigger convection, destabilising the proposed background state. The proposed SST is stated as

$$T_{SST}(\lambda, \phi) = \frac{T_{v,0} - \frac{u_0 a \Omega}{R_d} 4^n \left( F_3 - 2F_1 \right)}{1 + 0.608 q(\lambda, \phi, \eta = 1)}, \tag{16}$$

where $F_1$ and $F_3$ are described in Eq. (8).

To complete the surface initialisation, the Charnock parameter is specified to control the surface roughness (Charnock, 1955). The surface roughness lengths for momentum (M), heat (H) and total water (Q) air-surface transfers are defined in OpenIFS

(Eq. 3.26 ECMWF, 2017b) (and (Eq. 25 Beljaars, 1995)) as

$$z_{0M} = \alpha_M \frac{\nu}{u_\star} + \alpha_{Ch} \frac{u_*^2}{g} \tag{17a}$$

$$z_{0H} = \alpha_H \frac{\nu}{u_\star} \tag{17b}$$

$$z_{0Q} = \alpha_Q \frac{\nu}{u_\star}, \tag{17c}$$

where $\alpha_{Ch}$ is the Charnock parameter, $u_\star$ the friction velocity, $\nu$ kinematic viscosity, and $\alpha_M$, $\alpha_H$ and $\alpha_Q$ are constants set 0.11, 0.40 and 0.62, respectively. Being able to tune the Charnock parameter allows the modification of the surface friction which previous studies have shown to influence the intensity of extra-tropical cyclones (Adamson et al., 2006; Sinclair et al., 2010) and the structure of warm and cold fronts (Hines and Mechoso, 1993; Sinclair and Keyser, 2015).

## 2.4 Initial Perturbation

The baroclinic wave can be triggered by adding a localised unbalanced wind perturbation to a baroclinically unstable background state as the one described in Section 2.1. A Gaussian perturbation was chosen and it was centred at $(\lambda_c, \phi_c) = (\frac{\pi}{9}, \frac{2\pi}{9})$ which corresponds to 40° N, 20° E (Jablonowski and Williamson, 2006; Ullrich et al., 2015). The equation of the perturbation is given by

$$u_\epsilon(\lambda, \phi, \eta) = u_p \exp[-(\frac{r}{R})^2], \tag{18}$$

where $R = \frac{a}{10}$, $u_p = 10 \text{ ms}^{-1}$ and $r$ the great circle distance given by

$$r = a \arccos(\sin \phi_c \sin \phi + \cos \phi_c \cos \phi \cos(\lambda - \lambda_c)). \tag{19}$$

The final zonal wind field is obtained by adding $u_\epsilon$ to $u$ at each grid point at all model levels

$$u_{total}(\lambda, \phi, \eta) = u(\lambda, \phi, \eta) + u_\epsilon(\lambda, \phi, \eta). \tag{20}$$

## 3 Implementation in OpenIFS

### 3.1 OpenIFS

The proposed background state has been implemented in the Open Integrated Forecasting System (OpenIFS) cycle 43R3v2 (CY43R3) which is based on the Integrated Forecasting System of the European Centre for Medium-range Weather Forecasts (ECMWF) cycle 43R3, which was operational from July 2017 to June 2018 (ECMWF, 2017a). OpenIFS is a version of the Integrated Forecasting System model but does not include data assimilation capacities. Despite its name, OpenIFS is not open source but available to universities and research institutions under license. The model is hydrostatic, spectral and has the same physics parameterisation schemes as the full version of the IFS. In terms of applications, OpenIFS is able to compute deterministic and ensemble forecasts from either real, specified or idealised initial conditions. The project is coded in FORTRAN and in C, which is efficient for intensive and scientific computing.

## 3.2 Existing implementation

In OpenIFS CY43R3v2, the idealised background state implemented for the baroclinic wave test case is the one developed by Jablonowski and Williamson (2006). Originally, this background state was implemented in the full version of the IFS, and hence OpenIFS, to test the dynamical core. This background state was referred to as NTESTCASE 41 (dry case) and 42 (moist case) for the Dynamical Core Intercomparison Project (DCMIP). The original initial state of Jablonowski and Williamson (2006) has a very strong meridional temperature gradient which means that the near-surface temperature reaches -50°C at high latitudes. In the dry case with no physics parameterisation schemes, the surface heat fluxes are not computed meaning that the SSTs can be specified to be much warmer (or colder) than the near-surface atmospheric temperatures without causing any problems such as destabilisation of the boundary layer or convection. In contrast, in the moist case with physics parameterisation schemes turned on, exceptionally cold conditions at high latitudes with physically realistic SSTs cause large surface heat fluxes to develop and in the extreme case can result in low pressure centres resembling polar lows developing at high latitudes. Therefore, modifications to the Jablonowski and Williamson (2006) case are needed to enable it to be run with physics parameterisation schemes and to allow it to be used to investigate cyclone dynamics rather than the numerical accuracy of dynamical cores. Hence, the SST definition presented in Section 2.3 was used. Lastly, many aspects of the existing implementation are hard coded, and the parameters used to compute the background state were not accessible via the OpenIFS namelist.

## 3.3 The new implementation: OpenIFS baroclinic wave v1.0

The proposed background state is implemented into OpenIFS based on the derived analytical equations for the geopotential and temperature field as detailed in Section 2. Both fields contain non-trivial functions - such as the Gamma function (see Eq. (8)) - and can be difficult to implement. In order to avoid the costly use of factorials, $F_3$ was expressed as a binomial coefficient fraction and all the binomial coefficients were computed once with the multiplicative method, since $\binom{z}{k+1} = \frac{z-k}{k+1}\binom{z}{k}$ with $z$ and $k$ being integers. By using the definition for the gamma function for positive integers $z$

$$\Gamma(z) = (z-1)! \text{ and } \Gamma(z+1/2) = \frac{(2z)!}{4^z z!}\sqrt{\pi}, \tag{21}$$

the Gamma function can be replaced by binomial coefficients in $F_3$ as follows

$$
\begin{aligned}
\frac{\Gamma(k+n+3/2)}{\Gamma(k+n+2)} &= \frac{\Gamma(k+n+1+1/2)}{\Gamma(k+n+1+1)} \\
&= \frac{\Gamma(z+1/2)}{\Gamma(z+1)} \text{ where } z = k+n+1 \\
&= \frac{(2z)!}{4^z z!}\sqrt{\pi}\frac{1}{z!} \text{ Note: } \binom{2z}{z} = \frac{(2z)!}{z!z!} \\
&= \binom{2z}{z}\frac{\sqrt{\pi}}{4^z} \\
&= \binom{2(k+n)+2}{k+n+1}\frac{\sqrt{\pi}}{4^{k+n+1}}.
\end{aligned}
$$

**Table 1.** Template of the namelist used to set the proposed background state in the moist case. All other parameters under NAEPHY were set to false.

| Field | Parameter | Value | Explanation |
|---|---|---|---|
| NAMCT0 | N3DINI | 2 | Type of initial data, 2 = initial files ignored |
| NAMDYNCORE | LAPE | true | Aqua-planet simulation on/off (First trigger) |
| | LAQUA | true | Aqua-planet simulation on/off (Second trigger) |
| | MSSTSCHEME | 10 | Choice of SST forcing if aqua-planet enabled, no. 10 = eq. (16) |
| | NTESTCASE | 42 | Test case number for moist set-up |
| NAEPHY | LEPHYS | true | Master switch to enable physics parameterisation schemes on/off |
| | LEVDIF | true | Vertical diffusion on/off |
| | LESURF | true | Interactive surface processes on/off |
| | LECOND | true | Large scale condensation on/off |
| | LECUMF | true | Mass-flux convection scheme on/off |
| | LEPCLD | true | Prognostic cloud scheme on/off |
| | LEEVAP | true | Evaporation of precipitation on/off |
| | LEQNGT | true | Negative humidity fixer on/off |
| | LERADI | false | Radiation scheme on/off |
| | LERADS | false | Interactive surface radiative properties on/off |

$F_3$ can then be rewritten as

$$F_3 = \sum_{k=0}^{n} \binom{n}{k} (-1)^k \frac{1}{2(k+n)+1} \binom{2(k+n)+2}{k+n+1} \frac{\pi}{4^{k+n+1}}. \tag{22}$$

The proposed solution has been implemented as a new idealised case (indicated by the NTESTCASE parameter in OpenIFS), where the model state variables are initialised based on the equations for geopotential, virtual temperature, the horizontal wind components and, in the case of moist simulations, the specific humidity that were derived above. Once the initial values of the state variables are defined in the model, the OpenIFS simulations are integrated forward in time on an aquaplanet. In the moist case, most of the physics parameterisation schemes of OpenIFS are switch on, but the radiation parameterisation scheme and the wave model are deactivated. The customised SST function presented in Eq. (16) has been added to the other SST schemes already implemented in OpenIFS and is identified with the number 10 in the NAMDYNCORE namespace (variable name MSSTSCHEME in OpenIFS). In the OpenIFS namelist, the NAMDYNCORE and NAEPHY are important namespaces to fill in order to ensure the correct configuration and set up of the baroclinic wave simulation. The NAMDYNCORE namespace sets up all of the idealised model configurations and NAEPHY the different physics parameterisation schemes (i.e., whether they are activated or not). Finally, the NAMCT0 namespace set the main model control variables. N3DINI was set to 2 meaning that the meteorological values in the initial grib files were ignored and replaced by the idealised background state. The default values for the simulation in the moist case are presented in Table 1.

**Table 2.** Modifiable parameters with their default values and short description with units in NAMDIM namespace

| Parameter | Default value | OpenIFS given name | Function |
|---|---|---|---|
| $n$ | 3 | ZN | Jet width |
| $b$ | 2.0 | ZB | Jet height |
| $u_0$ | 35.0 | ZU0 | Together with $b$, $u_0$ adjusts the amplitude of zonal mean wind speed (ms$^{-1}$) |
| $T_{v,0}$ | 288.0 | ZT0 | Average surface virtual temperature (K) |
| $RH_0$ | 80.0 | ZRH0 | Surface level relative humidity (%) |
| $\gamma$ | 0.005 | ZGAMMA | Lapse rate (Km$^{-1}$) |
| $\alpha_{Ch}$ | 0.013 | ZCHAR | Charnock value |
| $u_p$ | 1.0 | ZUP | Amplitude of the zonal wind perturbation (ms$^{-1}$) |

In this version there is no decentering nor Asselin filter. The spectral diffusion used by default is of 4th order (with the exponent of the wavenumber dependency REXPDH=4) and is set to be rather weak, the strength of which is related to the used model timestep. The coefficients for $T_L 319$ are 2100.0 seconds for vorticity (HDIRVOR), divergence (HDIRDIV), temperature (HDIRT), humidity (HDIRQ) diffusions, and the other coefficients are set to zero.

### 3.4 How to use the new implementation?

Subsequent baroclinic wave simulations were run in the dry and moist case. The difference between the dry and moist case is the computation of virtual temperature (see Eq. (15)) and a non-zero specific humidity. In the dry case, the computation of specific humidity is disabled and thus the virtual temperature is equal to the real temperature at all time. The dry and moist test case can be computed by setting the NTESTCASE value to 41 or 42 respectively, replacing *de facto* the previous implementation. The current solution allows the user to switch on or off the perturbation specified in Section 2.4. Moreover, the user can define the amplitude of the Gaussian hill zonal wind perturbation by changing the value of ZUP in the namelist. It would be possible to use a perturbation with a different structure, but that would require the user to modify the source code. In total, six parameters were input to create various different background states and influenced the resulting baroclinic wave, one controlled the surface roughness ($\alpha_{Ch}$) and and one triggered the initial perturbation ($u_p$). All parameters were included in the NAMDIM namespace in the OpenIFS namelist. A default case was defined as shown in Table 2. Of these parameters, only ZCHAR and ZUP are not used to compute the initial background state.

A standalone version has been developed in FORTRAN and is available on Zenodo (https://doi.org/10.5281/zenodo.7890586). This standalone is divided in two parts: (1) a main program setting all the variables to compute the zonal fields and (2) a subroutine computing the zonal fields detailed in Sections 2.1 and 2.2. In Section 5, the "dry case" case refers to the dry simulations without physics parameterisation schemes and the "moist case" case is describing the case with physics parameterisation schemes and with moisture included.

## 4 Description of the experiments and diagnostics

This section is divided into four parts: (1) numerical stability of the initial background state (dry case and moist case), (2) meteorological stability and structure of the initial states, (3) temporal evolution of the default dry and moist baroclinic waves and (4) sensitivity of the evolution to different initial background states. All simulations are run at $T_L319$ L137 resolution (i.e., 63km horizontal resolution at the equator and 137 vertical levels with a model top of 0.01 hPa (https://confluence.ecmwf.int/display/UDOC/L137+model+level+definitions, accessed: 2023-12-05), with a timestep of 900 seconds for 15 days with an output frequency of 3 hours. Simulations without the perturbation were conducted to test the stability of the proposed background state: if implemented correctly and numerical errors are small, the initial state should not change in time. The jet width and height were varied by changing $n$ and $b$, and all other parameters presented in Table 2 were set to their default value. In total, 18 background states without the unbalanced wind perturbation were tested for 3 values of $n$ (1, 3, 6), 3 values of $b$ (1, 1.5, 2) and for both the dry and moist cases.

The Root-Mean-Square-Error (RMSE) is computed across all vertical levels for the zonal wind (Eq. 4.2, Jablonowski and Williamson, 2006) as

$$RMSE(u_{za}(t) - u_{ideal}(t=0)) \approx \left( \frac{\sum_{\eta_i=\eta_{surface}}^{\eta_{top}} \sum_{\phi_j=-90°}^{90°} [u_{za}(\phi_j,\eta_i,t) - u_{ideal}(\phi_j,\eta_i,t=0)]^2 w_{\phi_j} \Delta \eta_i}{\sum_{\eta_i=\eta_{surface}}^{\eta_{top}} \sum_{\phi_j=-90°}^{90°} w_{\phi_j} \Delta \eta_i} \right)^{\frac{1}{2}}, \tag{23}$$

where $u_{za}$ is the zonal average of the zonal wind speed, $u_{ideal}$ is the ideal zonal average of the zonal wind speed computed from the analytical expression for the zonal wind (Eq. (3)), $w_{\phi_j}$ is the weights to correct the convergence of the meridians $\phi_j$ and $\Delta\eta_i$ is the thickness of the model layer $\eta_i$.

Previous studies have used several metrics to test the meteorological stability of their background states such as the temperature, geopotential height, and zonal winds (Khairoutdinov et al., 2022), sometimes potential temperature, absolute vorticity and the Brunt-Väisälä frequency are added (Jablonowski and Williamson, 2006; Ullrich et al., 2015). The absolute vorticity, potential temperature, equivalent potential temperature (for the moist cases), zonal wind and Brunt-Väisälä frequency were computed to test if the initial state is stable to static (gravitational), inertial and symmetric instability. For the initial state to be absolutely stable to dry and saturated vertical displacements (static stability), equivalent potential temperature must increase with height everywhere. In the situation where equivalent potential temperature decreases with height, conditional instability is present, meaning that the atmosphere is stable to displacements of dry and unsaturated air parcels but unstable to displacements of saturated air parcels. If potential temperature decreases with height, then the atmosphere is absolutely unstable - both dry and saturated displacements are unstable. Thus, for the initial state to be absolutely stable potential temperature must increase with height and the Brunt-Väisälä frequency must be positive. Regions where equivalent potential temperature decreases with height are also stable and acceptable in the initial state as long as these regions are not saturated. For the initial state to be stable to horizontal displacements (inertial stability) the absolute vorticity must be positive (negative) in the northern (southern) hemisphere. Situations can exist where the atmosphere is statically and inertially stable, but the atmosphere is unstable to slantwise displacements (symmetric instability). This exists when the potential vorticity is negative. Hence for our initial state to be stable to inertial and symmetric instability, both the absolute vorticity and potential vorticity must be positive in the

northern hemisphere. However, although the initial state must be stable to static, inertial and symmetric stability, the set up must be baroclinically unstable. This requires the presence of a meridional temperature gradient in the mid-latitudes and a well defined zonal jet.

Several baroclinic waves (background states with perturbation) are generated for 6 values of $n$ (1 to 6) and the default values for the remaining parameters (presented in Table 2). All Baroclinic Wave Simulations (BWS) have been run for 15 days. During the 15 day simulation, the first cyclone to emerge develops directly from the initial perturbation, however, up-stream and downstream development also occurs resulting in multiple cyclones and anticyclones. To objectively identify the centre of the cyclone which develops directly from the initial perturbation (the first cyclone) the TRACK software (Hodges, 1994, 1995, 1999) is used. Cyclones are identified as localised maxima in the 850-hPa relative vorticity field truncated to T42 spectral resolution. Only tracks lasting for 4 days and which travel 1000 km are retained. The weak tracks are removed by setting a threshold of $0.4 \times 10^{-5} \mathrm{s}^{-1}$ for the T42 850-hPa relative vorticity. This threshold is weaker than the threshold usually applied when using TRACK ($1 \times 10^{-5} \mathrm{s}^{-1}$) to identify cyclones in the real world to enable the first cyclone to be detected as soon as possible.

## 5 Results

### 5.1 Numerical stability of the initial background state

As shown in Figure 3, the RMSE for all of the dry cases with no perturbation included are below $0.005 \mathrm{~ms}^{-1}$ with no noticeable increase over time. These values are lower than the RMSE in the Jablonowski and Williamson (2006) report for the same semi-Lagrangian setup and lower than the RMSE reported by Khairoutdinov et al. (2022). This means that the specified background states are stable and correctly implemented into OpenIFS. A slight increase in RMSE over time can be observed in the moist cases, however, in comparison to the dry cases the RMSE in the moist cases is at most 6% larger by the end of the 15 day simulation. The lower the $n$ value and the higher the $b$ value, the higher the RMSE. This tendency can be understood as: the more extensive the jet is, the higher the RMSE.

### 5.2 Meteorological stability and structure of the initial background state

The initial fields for the default values ($n$=3 and $b$=2.0) in the moist case are presented in Figure 4. The initials fields for the default values in the dry case are presented in the Supplementary material (Figure S4) as they are very similar to the moist case with the obvious exceptions of the specific humidity (zero everywhere in the dry case) and the equivalent potential temperature. In both the dry and moist cases, a strong baroclinic zone is present between 30 and 60 degrees (Figure 4 a, c and Figure 5 d) with temperatures at the surface in the moist (dry) case reaching a maximum of 22.6°C (26.0°C) at the equator and decreasing to -13.7°C (-12.8°C) at the poles. These temperatures are similar to the temperatures reported in the ERA5 global re-analysis, which display a surface temperature between 20°C and 30°C in the tropics and between -10°C and -20°C at the poles (Hersbach et al., 2020). The baroclinic zone is co-located with a jet stream between 20°N/S and 70°N/S

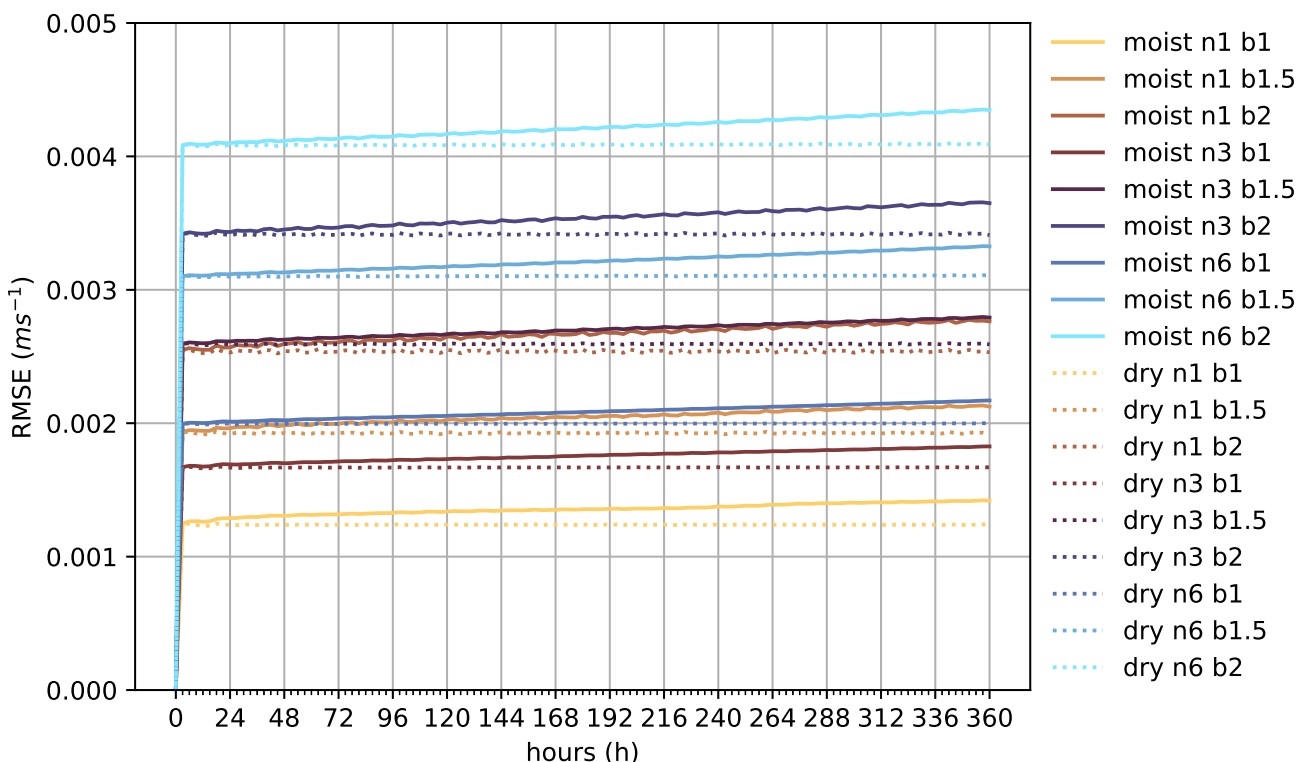

**Figure 3.** RMSE as a function of time for all cases. The solid lines represents the moist cases and the dotted lines the dry cases. The colour map is from Crameri et al. (2020).

with a maximum zonal wind speed of 30 ms$^{-1}$ (identical in the dry case) located in the centre of the jet at 45°N/S and
250 hPa as shown in Figure 4 a, c. When the width and height of the idealised jet streams that we propose in this study
are compared to realistic values from ERA5 (see Lee et al. (2023)), reasonable agreement is found with the climatological
winter means over the northwestern Atlantic. However, the maximum zonal wind speed is on the lower range of the maximum
wind speed of the ERA5 profile, which is not a problem considering that $u_0$ is a tunable parameter as presented in Table 2.
Moreover, the December mean zonal wind speed over the years 1979 - 2021 was computed (Figure S1) and compared to our
idealised jet structures (Figure S2). A 10° location difference of the centre of the northern jet have been found but the proposed
analytical solution is close in shape and intensity to this profile. The absolute vorticity, presented in Figure 4 b, demonstrates
that the default initial state is inertially stable. The condition for symmetric stability, that potential vorticity is positive in the
northern hemisphere and negative in the southern hemisphere, is also met. This can be inferred from the potential temperature
distribution (Figure 4 c) combined with the absolute vorticity distribution (Figure 4 b). The potential temperature (Figure 4 c)
increases with height everywhere in the model domain indicating that the initial state is stable to dry static stability. Figure 4 e
shows that the equivalent potential temperature increases everywhere, except for the tropical regions. This means that the initial

state is stable to moist static stability, as long as air parcels are not saturated in the tropics. The static stability is also indicated by the Brunt-Väisälä frequency presented in (Figure 4 d). Relatively high values for the Brunt-Väisälä frequency were seen at high latitudes at most pressure levels, indicating high static stability. A lower Brunt-Väisälä frequency and thus lower static stability was observed around the equator and near the surface in the polar regions.

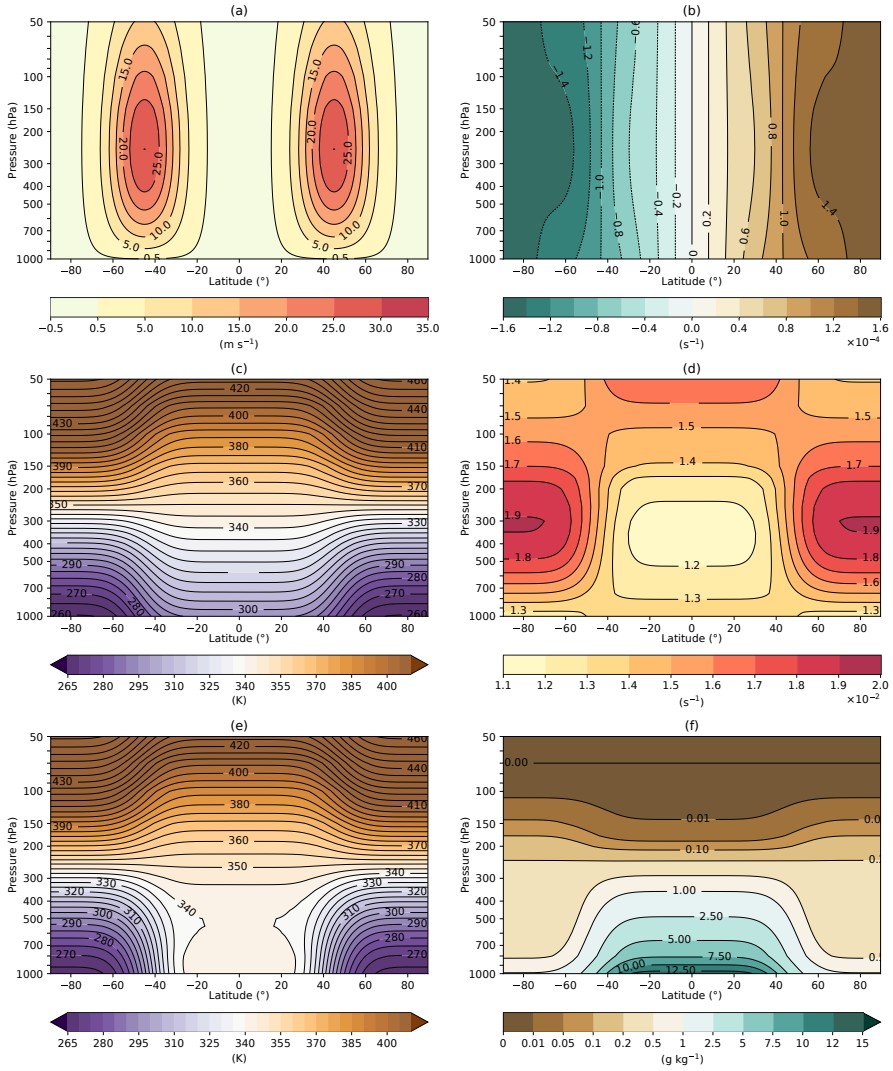

**Figure 4.** Cross sections of the default moist initial background state ($n$=3 and $b$=2.0) for: (a) zonal wind speed, (b) absolute vorticity, (c) potential temperature, (d) Brunt-Väisälä frequency, (e) equivalent potential temperature and (f) specific humidity.

Figure 5 shows the initial temperature, zonal wind speed and dynamical tropopause (taken here to be the 2 PVU surface) for different values of $n$ (1, 3 and 6) and $b$ (1, 1.5 and 2), with Fig. 5d showing the default initial state which was presented in Fig. 4. The same figure has been produced for the dry cases (Figure S5) and can be found in the Supplementary material, but this figure is not included here because of the similarity with Figure 5. Increasing $n$ causes the meridional width of the jet to decrease. In the case of $n = 1$, the jet is very wide extending from 15°N/S to 75°N/S. In contrast, when $n = 6$, the jet is constrained between 30 and 60°N/S. Increasing $n$ also causes the width of the baroclinic zone to decrease and the surface temperature gradient between the equator and pole to decrease: in the case of $n = 1$, the temperature difference between the equator and pole is 57.3°C whereas for $n = 6$ the temperature difference is 26.3°C (for $b$=2.0). Decreasing $b$ leads to the displacement of the centre of the jet toward the surface. For all initial states, higher values for $n$ lead to a narrower jet and a stronger temperature gradient (Figure 5). Moreover, high values of $b$ lead to a stronger temperature gradient, a higher centre of the jet and an increase in the maximum wind speed of the jet as shown by the innermost contour (e.g., Figure 5 a and c). Additionally, the centre of the jet gets closer to the surface with decreasing $b$. For $b$=1, the centre of the jet is lower than the dynamical tropopause at 2 PVU, but does not resemble a realistic jet structure as found in reanalysis ((Lee et al., 2023) and Figure S1 and S2 in the Supplementary material). Thus, when $b$=1, the jet's width is unrealistic meaning that it is not recommended to use $b$=1 when studying extra-tropical cyclone dynamics. The latitudinal location of the centre of the jet is independent of $n$ and $b$ and located at 45° in both hemispheres for all the different initial cases presented. The height of the dynamical tropopause increases with increasing $n$, which means that the static stability increases with higher $n$ (Held, 1982).

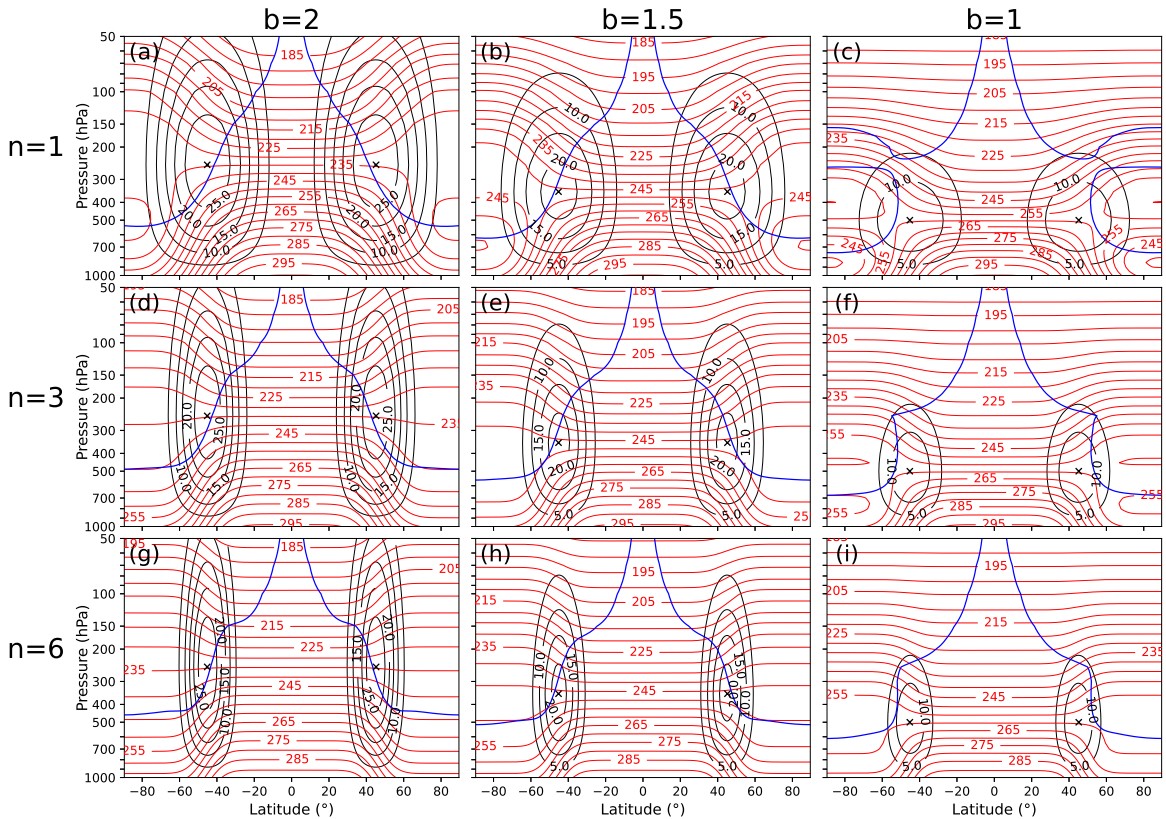

**Figure 5.** The zonal wind speed in $\mathrm{ms}^{-1}$ (black contours), temperature in K (red contours) and dynamical tropopause at 2 PVU (blue contours) fields of the moist initial state for different values of $n$ and $b$.

Based on the results of the Sections 5.1 and 5.2, it is evident that the initial background states are stable through time and balanced from a meteorological perspective. As shown in Figure 5, the background states display strong baroclinicity in the mid-latitudes. The next section will present the evolution of the baroclinic waves triggered in some of the presented background states ($b$=2.0).

## 5.3 Temporal evolution of the dry and moist default baroclinic waves

### 5.3.1 Dry case

The evolution of the dry default baroclinic wave ($n$=3, $b$=2), which is also run with no physics parameterisation schemes switched on, is shown in Figure 6. The formation of a very weak closed low pressure system is just evident after 144 h

(Figure 6a). At this time, the minimum MSLP is located slightly poleward of 45°N which was the central latitude of the initial perturbation. Also at 144 h, a small amplitude wave in the 850-hPa temperature is evident co-located with the developing cyclonic circulation but pronounced fronts have not yet developed. After 168 h of development (Figure 6b), the cyclone has a minimum MSLP of 1006.75 hPa, 6.5 hPa lower than the initial surface pressure. Two areas of high pressure are evident on either side of this cyclone and another cyclone has begun developing upstream of the cyclone which developed directly from the initial unbalanced perturbation. By 192 h, (Figure 6c) a well developed cyclone is present with a minimum MSLP of 1000.8 hPa. Cold and warm fronts are now evident in the 850-hPa temperature. The centre of the cyclone has continued to move polewards and the two anticyclones on either side have also intensified slightly and moved equatorwards. In addition to the upstream development that was evident 24 h previously, downstream development has also started to take place by 192 h; a third low pressure centre / cyclone is now visible at 150°E although this is the weakest of all three cyclones. Rapid intensification takes place and by 228 h (Figure 6d), the initial cyclone has deepened considerably (minimum MSLP of 983.3 hPa, and almost 30 hPa lower than the initial surface pressure). Cold and warm fronts with large thermal gradients are present and an occlusion has begun to develop. Furthermore at 228 h, both the upstream and downstream cyclones have continued deepening, with the upstream cyclone being deeper yet spatially smaller than the downstream cyclone. Furthermore, the horizontal distance between the upstream cyclone and the initial cyclone is smaller than between the downstream cyclone and the initial cyclone. By 264 h (Figure 6e), the initial cyclone is very mature, deep, has a clear warm seclusion, and has moved farther polewards. The downstream cyclone has undergone notable deepening since 228 h and a fourth, relatively small cyclone has started to develop upstream at 60°E. After 312 h of simulation, a much more chaotic picture is evident. The first upstream cyclone has merged with the trailing cold front of the initial cyclone leading to the presence of a very large cyclonic system which has a very strong north-south pressure gradient on its southern flank. The downstream cyclone has continued intensifying and has moved polewards leading to a very meridionally extended system that has strong fronts associated with it. The upstream cyclone has also continued to intensify but remains weaker and smaller in spatial scale than the other cyclones.

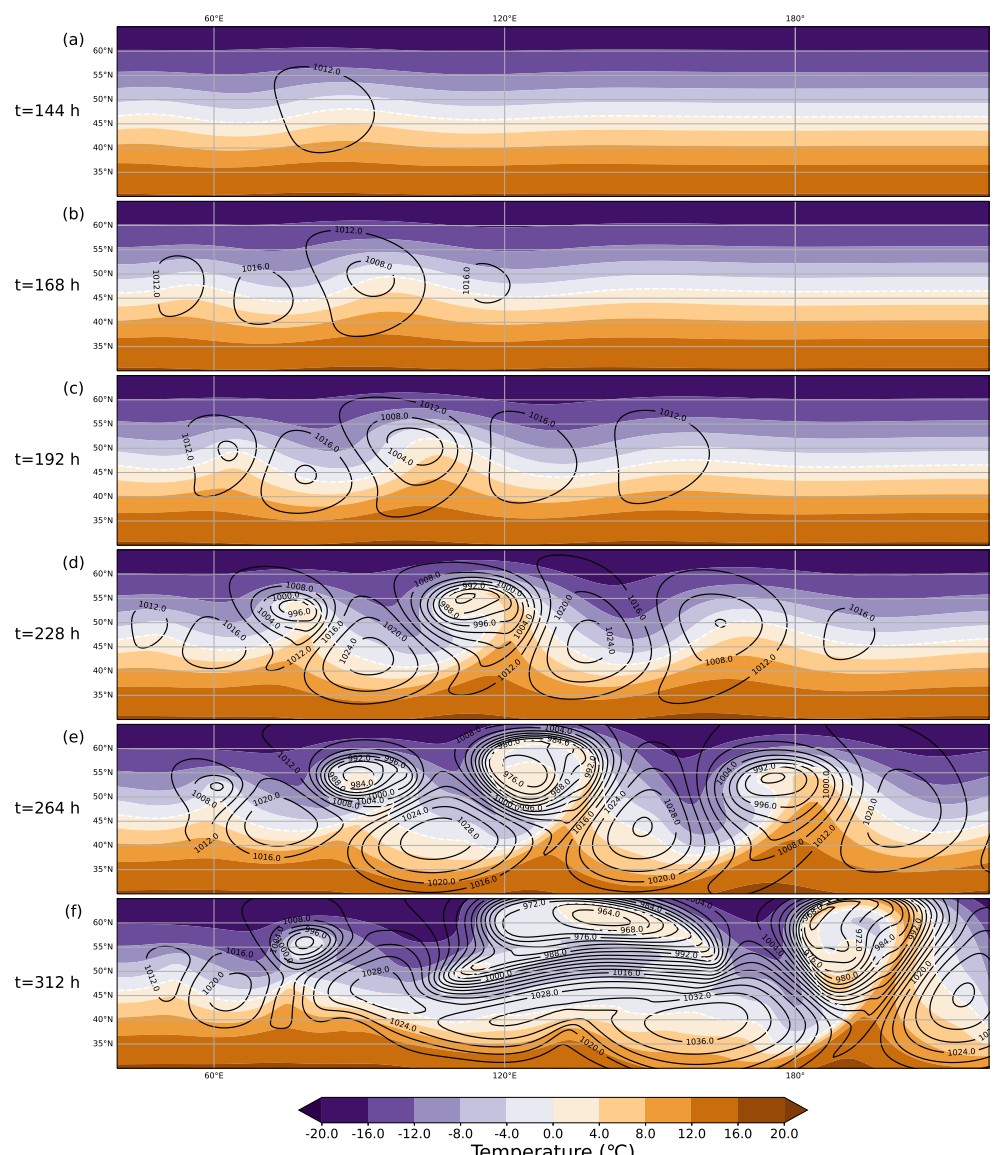

**Figure 6.** The development of the baroclinic wave for the default dry scenario ($n=3$ and $b=2.0$) at (a) t=144 h, (b) t=168 h, (c) t=192 h, (d) t=228 h, (e) t=264 h, (f) t=312 h. The black contours show mean sea level pressure (hPa), the shading shows the temperature (°C) at 850 hPa. Note: this figure does not show the whole model domain; the x-axis ranges from 40-220°E, while the y-axis ranges from 30-65°N.

### 5.3.2 Moist case

The evolution of the default baroclinic wave (n=3, b=2) with moisture is shown in Figure 7 and the associated precipitation patterns in Figure 8. After 144 h of development (Figure 7a), the structure of the moist baroclinic wave is very similar to the dry case at the same time and only a very weak cyclonic circulation and no precipitation has developed at this time (Figure

8a). At 168 h (Figure 7b), the initial cyclone has deepened to 1007.5 hPa which is slightly weaker than the dry case. A thermal wave is also evident at 168 h and an upstream cyclone has begun to develop but no precipitation has developed at this stage (Figure 8b). At 192 hours, the general evolution of the moist baroclinic continues in a similar manner to the dry case. A well

developed cyclone is now present with a minimum MSLP below 1003 hPa, which is again slightly higher than in the dry case. Cold and warm fronts are now evident in the 850-hPa temperature and precipitation is now evident on the warm front of the initial cyclone and, to a lesser extent, in the developing warm sector of the upstream cyclone. Rapid development takes places between 192 h and 228 h and by 228 h, a strong cyclone with clear fronts is evident. Furthermore, by 228 h moderate-to-heavy precipitation (values exceeding 3 mm per 3 hours) is present along the warm fronts of the initial cyclone and the upstream

cyclone as well as on the warm side of the cold front and in the warm sector these two cyclones. In the moist case, after 264 h, three well developed cyclones are evident, all of which have a surface pressure trough co-located with the cold front (Figure 7e). In comparison to the dry case, the moist case has much weaker horizontal pressure gradients particularly on the the southern and northern sides of the initial cyclone. Precipitation now covers a larger area than 24 hours earlier and extends further south along the cold fronts of each of these three well developed cyclones (Figure 8e). The second and weaker upstream

cyclone also produces precipitation at this stage. After 312 h of development, the initial cyclone and the first upstream cyclone have become very mature systems. The strongest cyclone at this stage is the downstream cyclone which has a minimum mean sea level pressure below 972 hPa (41 hPa lower than the initial surface pressure). Precipitation is still associated with all 4 cyclones (Figure 8f). The downstream cyclone has the heaviest and most expansive area of precipitation whereas the first upstream cyclone only has moderate precipitation remaining on the trailing cold front. In comparison to the dry case at 312

440   h, all cyclones in the moist case are weaker and less developed and the first upstream and initial cyclone remain as separate features.

The slower rate of development in the moist case is very likely caused by the presence of surface friction (and other physics parameterisation schemes) in the moist simulations whereas the dry cases are run with no friction and no parameterised physics schemes. This result is in agreement with Boutle et al. (2010) who, in idealised baroclinic life cycle simulations found that a

deeper cyclone developed in their dry, no boundary layer case (i.e., no boundary-layer scheme nor surface friction was included in the simulations) than in their moist simulation in which the boundary layer scheme was activated.

Overall, Figures 6, 7 and 8 demonstrate that the cyclones and anticyclones that develop from the default initial state are realistic and consequently can be used to investigate cyclone dynamics. In comparison to previous BWS studies that have used limited model domains with periodic boundaries (Feldstein and Held, 1989; Hoskins et al., 1977), the set up presented here

and the evolution of the default baroclinic wave allow for studies into both upstream and downstream development.

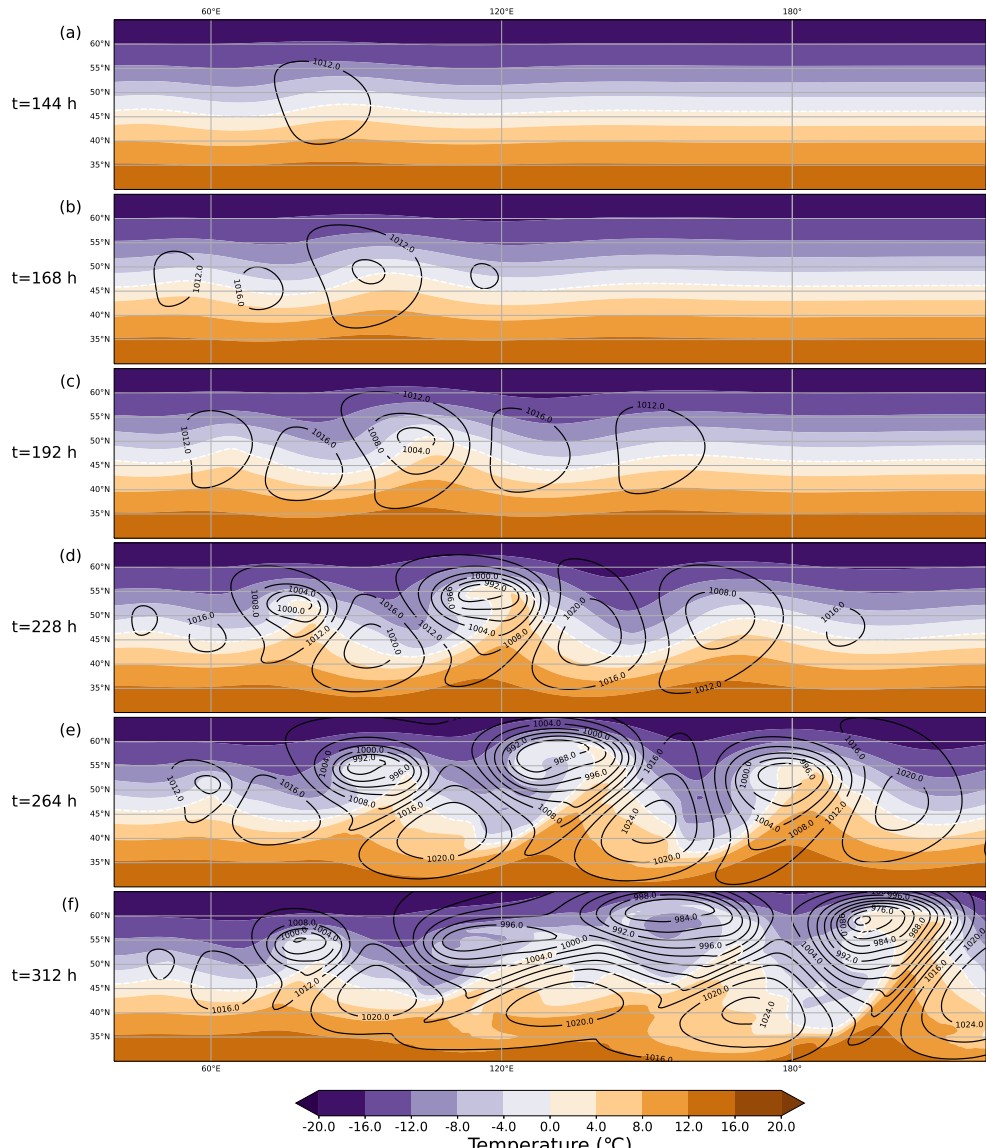

**Figure 7.** The development of the baroclinic wave for the default moist scenario ($n$=3 and $b$=2.0) at (a) t=144 h, (b) t=168 h, (c) t=192 h, (d) t=228 h, (e) t=264 h, (f) t=312 h. The black contours show mean sea level pressure (hPa), the shading shows the temperature (°C) at 850 hPa. Note: this figure does not show the whole model domain; the x-axis ranges from 40-220°E, while the y-axis ranges from 30-65°N.

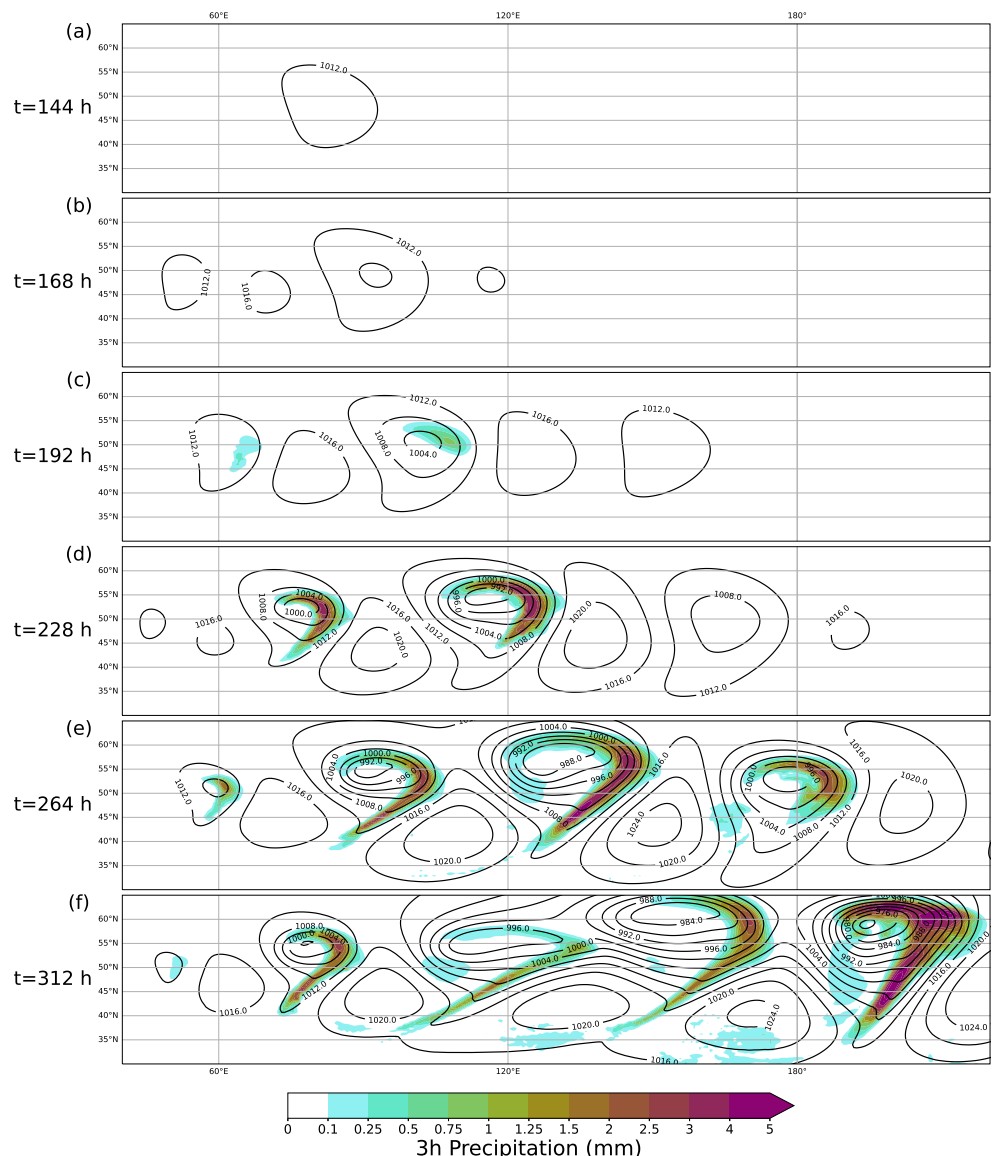

**Figure 8.** The development of the baroclinic wave for the default moist scenario ($n=3$ and $b=2.0$) at (a) t=144 h, (b) t=168 h, (c) t=192 h, (d) t=228 h, (e) t=264 h, (f) t=312 h. The black contours show mean sea level pressure (hPa), the shading shows the 3h precipitation (in mm). Note: this figure does not show the whole model domain; the x-axis ranges from 40-220°E, while the y-axis ranges from 30-65°N.

## 5.4 Impact of different initial states on the baroclinic wave evolution

Figures 9 and 10 demonstrate how the structure of the dry and moist baroclinic wave depend on the value of $n$ after 204 h. In both the dry and moist case, decreasing $n$ increases the width of the jet stream and the baroclinic zone (Fig. 5), which means that the resultant cyclones have a greater meridional extent with smaller values of $n$ (Figs. 9, 10). Decreasing $n$ leads

to a reduction in the minimum surface pressure of the cyclones in both the dry and moist cases. In the dry case, for n=1, the minimum surface pressure of the first developing cyclone after 204 h is 989 hPa (Fig. 9a) whereas for n=6 the corresponding value is 1000 hPa (Fig. 9f). The decrease in surface pressure with increasing $n$ also holds true for the cyclones which develop upstream and downstream and is likely because the 850-hPa temperature difference between the equator and the pole, and hence available potential energy and baroclinicity, is larger with smaller $n$ (Fig. 10). Decreasing $n$ also causes the phase speed

of the cyclones to increase and the cyclones and anticyclones travel eastward faster with larger $n$, despite that the maximum speed of the zonal jet does not change considerably. This increase in phase speed is observed in both the dry and moist cases. A detailed investigation of how all parameters which control the structure of the initial state (as presented in Table 2) affect the intensity and structure of the resultant cyclones will be the topic of a future study.

Figure 11 shows the temporal evolution of the maximum 850-hPa relative vorticity as identified by TRACK of the cyclone

which develops first in both the dry and moist experiments with different $n$. Note that this is not necessarily the cyclone which experiences the largest deepening rates nor the largest maximum vorticity but it is the cyclone which is directly caused by the initial unbalanced perturbation. The first cyclone in all experiments is detected by TRACK after 5 – 6 days. Between day 7 and day 10, a period of rapid intensification occurs for all experiments. In both the dry and moist cases, as $n$ decreases, the rate of intensification increases particularly between day 8 and day 10, and the eventual maximum value of relative vorticity is

larger for smaller $n$. This is consistent with Figs. 9 and 10 which showed that the minimum surface pressure was also lower with smaller $n$ and that the 850-hPa temperature difference between the poles and the equator was larger with smaller $n$. Furthermore, increasing $n$ results in the maximum vorticity being reached at an earlier time in both the dry and moist cases. Figure 11 also clearly highlights the difference between the moist and dry simulations. The maximum vorticity of the first developing cyclone is larger and occurs later in time in the dry cases compared to the corresponding moist cases.

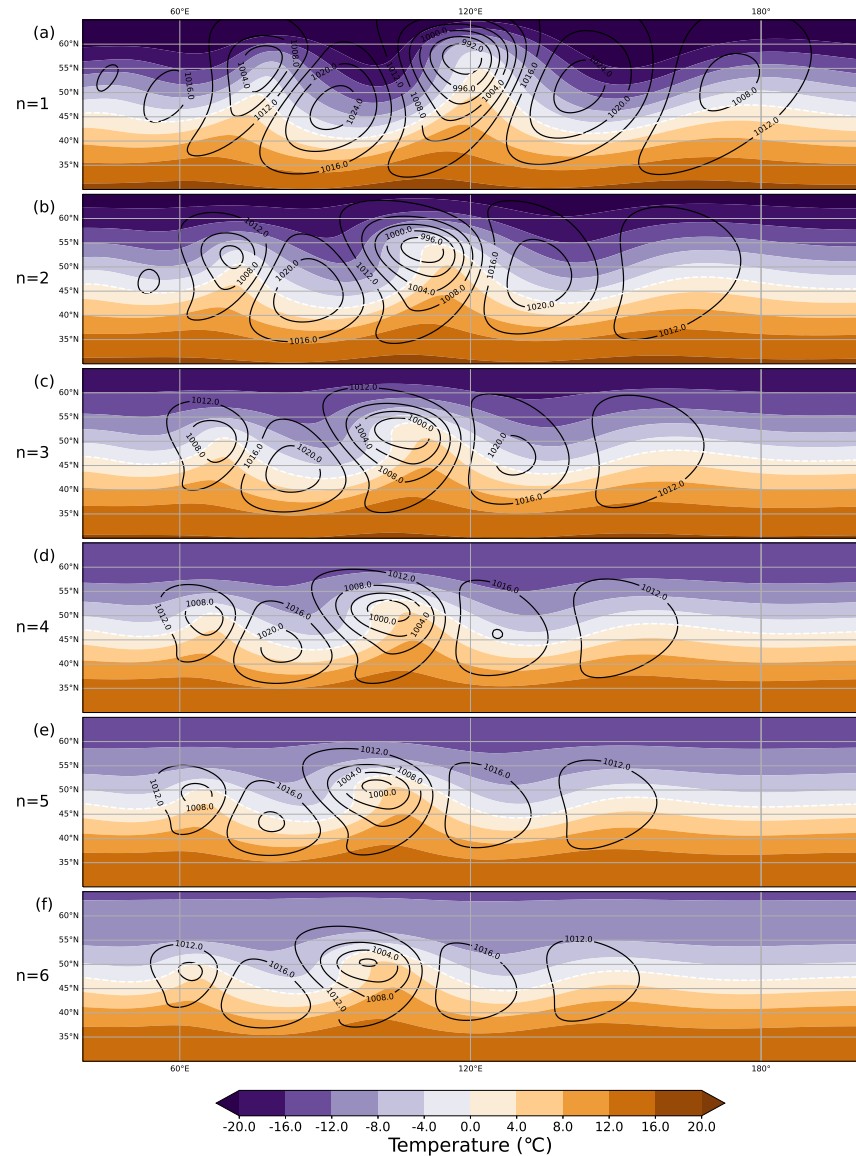

**Figure 9.** The development of the baroclinic wave at t=204h for (a) n=1, (b) n=2, (c) n=3, (d) n=4, (e) n=5, (f) n=6 in the dry case (for fixed *b*=2.0). The black contours show mean sea level pressure (hPa), the shading shows the temperature (°C) at the 850 hPa level. Note: this figure does not show the whole model domain; the x-axis ranges from 40-220°E, while the y-axis ranges from 30-65°N.

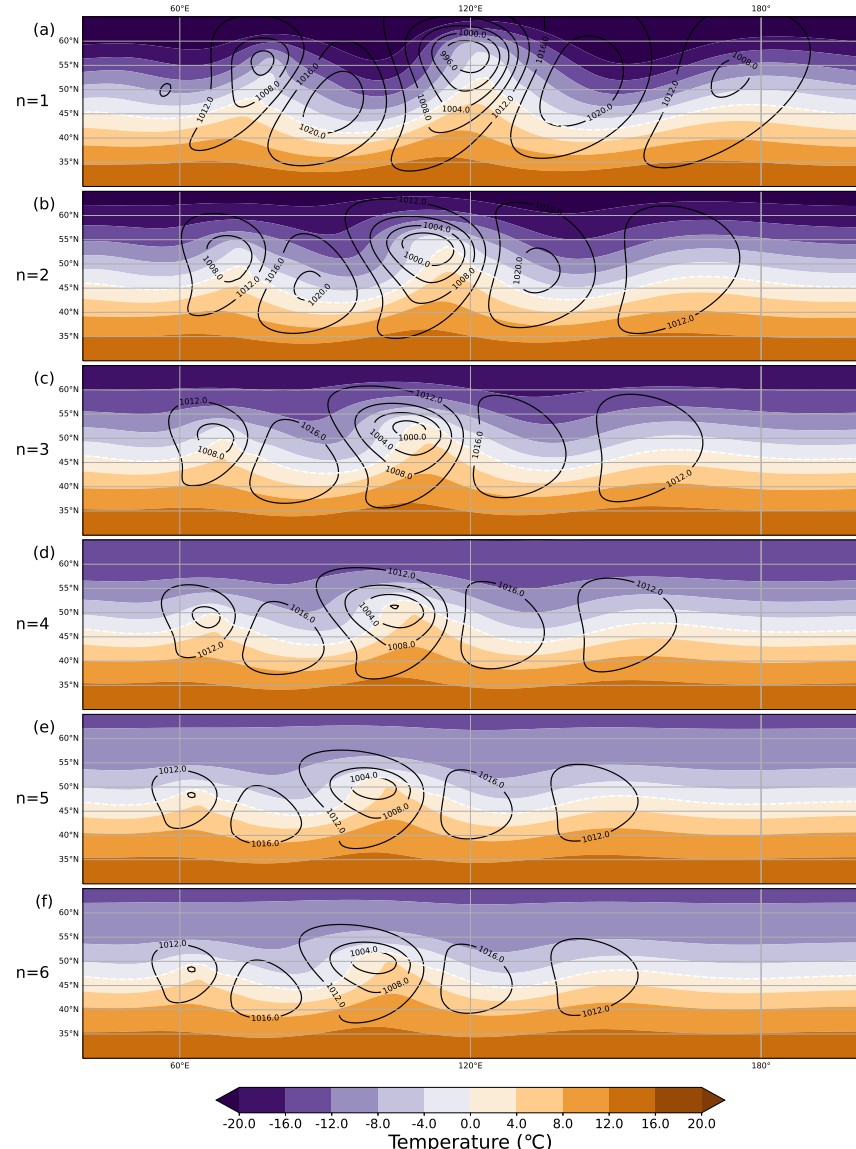

**Figure 10.** The development of the baroclinic wave at t=204h for (a) n=1, (b) n=2, (c) n=3, (d) n=4, (e) n=5, (f) n=6 in the moist case (for fixed $b$=2.0). The black contours show mean sea level pressure (hPa), the shading shows the temperature (°C) at the 850 hPa level. Note: this figure does not show the whole model domain; the x-axis ranges from 40-220°E, while the y-axis ranges from 30-65°N.

## 475 6 Conclusions

This article introduced idealised initial background states for a baroclinic lifecycle simulations. The main advantages of these background states are that they can entirely be expressed analytically and controlled through configuration files. The jet struc-

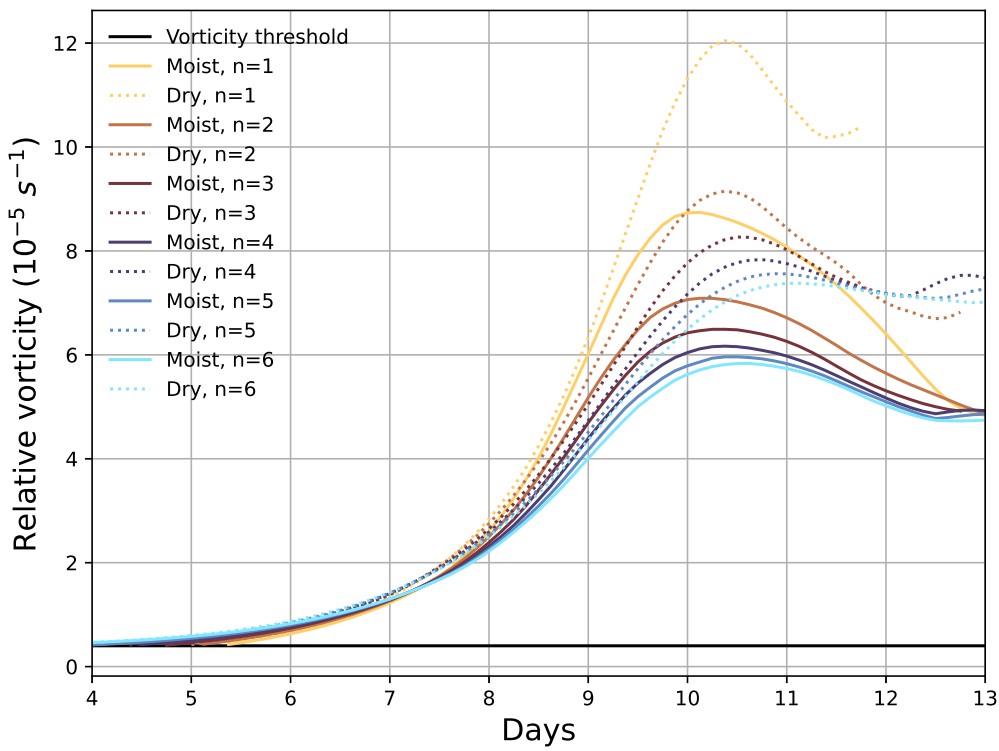

**Figure 11.** Evolution of the maximum relative vorticity of the first cyclone to develop as part of the baroclinic wave as a function of time. Solid lines represent the moist cases and the dashed lines the dry cases (for fixed $b$=2.0). The black horizontal line shows the threshold for relative vorticity set to $0.4 \ 10^{-5}\mathrm{s}^{-1}$ used by the tracking algorithm TRACK. The colour map is from Crameri et al. (2020).

ture and strength can be tuned as well as the average virtual temperature, the surface relative humidity, the lapse rate and the surface roughness. This flexibility allows an easy generation of different background states and their related baroclinic waves.

All studied initial background states are proven to be stable even in the moist case scenario. Moreover, a Gaussian perturbation of the zonal wind speed allows the development of a baroclinic wave in the dry and moist cases, which depends on the given jet structure. The presented solution is appealing for two main reasons.

    First, the proposed solution is implemented in OpenIFS CY43R3, which is a popular state-of-the-art model used extensively for meteorological and climate research (Carver, 2022). The idea was to propose a solution easy to test and use it to allow a

wide accessibility. Moreover, the Appendix presents the exhaustive integration and derivation of the initial background state to allow easy modification of the analytical formulae of the zonal wind speed field and the easy identification of the potential difficulties of its integration. One limitation of our initial state is that in its current form it is not possible to easily add barotropic shear to the low-level of the jet. Barotropic shear has been proven to be a defining condition for the structure development of the

extra-tropical cyclones (Agustí-Panareda et al., 2005). Future work will address this issue by proposing an analytical solution including an analytical barotropic shear.

Second, even if the initial background states may be unstable to moist saturated air parcel displacements, for our default simulation of n=3, b=2, no convective available potential energy (CAPE) is present in the tropics (0°N). This shows that even when the air parcel at the surface is lifted, it will stay cooler than its environment, meaning that the atmosphere is stable. In the case of n=1 b=1, the CAPE is large, due to a stronger decrease in temperature with height in the troposphere. However, as this CAPE can only be released once the surface parcel reaches saturation or is substantially lifted, and the fact that this area is not affected by the baroclinic wave, this does not influence the meteorological stability of our setup. The initial background states are proven to be realistic considering ERA5 average temperature and zonal wind speed cross-sections. Moreover, this study has proven the sensitivity of the BWS to the height and width of the jet, which has been possible by the control of the different parameters through OpenIFS namelist. The proposed solution is of interest to create a large ensemble of baroclinic lifecycles which would allow the study of the sensitivity of ETC to various initial background states. Future study will investigate the dependency of several measures of cyclone's intensities - such as mean sea level pressure, relative vorticity at 850 hPa, 10-m wind gust and SSI (Leckebusch et al., 2008) - to the initial conditions of cyclogenesis.

*Code and data availability.*  The licence for using the OpenIFS model can be requested from ECMWF user support (openifs-support@ecmwf.int). The modified subroutines of OpenIFS, a standalone version to compute the zonal fields detailed in Sections 2.1 and 2.2, the submission and plotting scripts, the configuration files and the raw data are available on Zenodo (https://doi.org/10.5281/zenodo.7890586).

## Appendix A:  Detailed Derivation of Analytical Initial Conditions

The derivation of the analytical initial conditions start from the primitive equations for moist adiabitic and frictionless flow in spherical coordinates $(\lambda, \phi)$ and vertical $\eta$ levels by the equations for

u-momentum: $\dfrac{du}{dt} - \dfrac{uv\tan\phi}{a} = -\dfrac{1}{a\cos\phi}\left(\dfrac{\partial\Phi}{\partial\lambda} + R_d T_v \dfrac{\partial\ln p}{\partial\lambda}\right) + fv$  (A1a)

v-momentum: $\dfrac{dv}{dt} + \dfrac{u^2\tan\phi}{a} = -\dfrac{1}{a}\left(\dfrac{\partial\Phi}{\partial\phi} + R_d T_v \dfrac{\partial\ln p}{\partial\lambda}\right) - fu$  (A1b)

hydrostatic balance: $\dfrac{\partial\Phi}{\partial\eta} = -\dfrac{R_d T_v}{p}\dfrac{\partial p}{\partial\eta}$  (A1c)

continuity: $\dfrac{\partial}{\partial t}\left(\dfrac{\partial p}{\partial\eta}\right) + \dfrac{1}{a\cos\phi}\dfrac{\partial}{\partial\lambda}\left(u\dfrac{\partial p}{\partial\eta}\right) + \dfrac{1}{a\cos\phi}\dfrac{\partial}{\partial\phi}\left((v\cos\phi)\dfrac{\partial p}{\partial\eta}\right) + \dfrac{\partial}{\partial\eta}\left(\dot{\eta}\dfrac{\partial p}{\partial\eta}\right) = 0,$  (A1d)

thermodynamic: $\dfrac{dT}{dt} - \dfrac{R_d T_v \omega}{c_p p} = 0$ and  (A1e)

moist ideal gas law: $p = \rho R_d T_v,$  (A1f)

where $d/dt$ is the full time derivative in spherical coordinates given as

$$\frac{d}{dt}() = \frac{\partial}{\partial t} + \frac{u}{a\cos\theta}\frac{\partial}{\partial\lambda}() + \frac{v}{a}\frac{\partial}{\partial\phi}() + \dot{\eta}\frac{\partial}{\partial\eta}().$$

Please note that the specific heat capacity of air, $c_p$, in the thermodynamic equation needs to be corrected with a correction factor $\delta$ when moisture is included in the model. The correction factor is defined as

$$\delta = 1 + (c_{p,vap}/c_{p,dry} - 1)q,$$

where $c_{p,vap}/c_{p,dry}$ = 1860/1004 (units: J/(kg K)) and $q$ is the specific humidity (units: kg/kg) (ECMWF, 2017a, eq. (2.3)).

The following equalities are used in the derivations of the geopotential field

$$* \sin(2\phi) = 2\sin\phi\cos\phi$$

$$** \sin^2\phi = 1 - \cos^2\phi$$

$$*** (a+b)^n = \sum_{k=0}^{n}\binom{n}{k}a^{n-k}b^k$$

$$+ \tan\phi = \frac{\sin\phi}{\cos\phi}$$

$$\dagger \int_{-\pi/2}^{\pi/2} \cos^{2(k+n+1)}\phi\, d\phi = \sqrt{\pi}\frac{\Gamma(k+n+3/2)}{\Gamma(k+n+2)} \text{, see e.g. WolframAlpha}$$

$$\ddagger \int_{-\pi/2}^{\pi/2} \sin^{2(k+2n+1)}\phi\cos\phi\, d\phi = \int_{-1}^{1} u^{2(k+2n+1)}du = \frac{2}{2(2n+k+1)+1}.$$

Whenever an equality is used its symbol is noted above the equality sign, for example $\overset{*}{=}$.

## A1  Mean Geopotential Field

The mean virtual temperature field is defined as

$$\langle T_v \rangle = T_{v,0} - \gamma z, \tag{A2}$$

where $T_{v,0}$ is the reference virtual temperature, $\gamma$ the lapse rate and $z$ height (moist version of (Eq. (10), Ullrich et al., 2015)). The geopotential mean field is derived from the mean virtual temperature field and the hydrostatic balance eqauation as follows

$$\frac{\partial p}{\partial z} = -\rho g$$

$$\frac{\partial p}{\partial z} = -\frac{p}{R_d\langle T_v\rangle}|\text{ Eq. (A2)}$$

$$\frac{\partial p}{p} = -\frac{g}{R_d(T_{v,0}-\gamma z)}\partial z|\text{ integrate LH and RH}$$

$$\int_{p_s}^{p}\frac{\partial p}{p} = -\frac{g}{R_d}\int_{z_0=0}^{z}\frac{1}{(T_{v,0}-\gamma z)}\partial z. \tag{A3}$$

The left-hand side of Eq. (A3) is solved as

$$\int_{p_s}^{p} \frac{\partial p}{p} = \ln \frac{p}{p_s}, \tag{A4}$$

and the right-hand side of Eq. (A3) is solved by substitution of $u = T_{v,0} - \gamma z$. The right-hand side is solved as

$$-\frac{g}{R_d} \int_{z=0}^{z} \frac{1}{(T_{v,0} - \gamma z)} \partial z = \frac{g}{R_d \gamma} \ln \frac{Tv,0 - \gamma z}{T_{v,0}}. \tag{A5}$$

By solving for $z$, this gives us that

$$z = \frac{T_{v,0}}{\gamma} [1 - (\frac{p}{p_s})^{\frac{R_d g}{\gamma}}], \tag{A6}$$

and by realising that $p/p_s$ is $\eta$, the mean geopotential field is then

$$\langle \Phi \rangle = \frac{g T_{v,0}}{\gamma} (1 - \eta^{\frac{R_d g}{\gamma}}). \tag{A7}$$

## A2  Geopotential Field

The geopotential anomaly field is derived from the steady-state momentum equation for $v$ with the zonal flow defined by $u = -u_0 \ln \eta \exp(-[\ln \eta / b]^2) \sin^{2n} 2\phi$

$$\frac{1}{a} \frac{\partial \Phi'}{\partial \phi} = -(-u_\eta \sin^{2n} 2\phi) \left( 2\Omega \sin \phi + \frac{-u_\eta \sin^{2n} 2\phi}{a} \tan \phi \right), \tag{A8}$$

where $u_\eta = u_0 \ln \eta \exp(-[\ln \eta / b]^2)$. Equation (A8) is multiplied by $a$ and then integrated analytically over $\phi$

$$\Phi'(\lambda, \phi, \eta) = \int \left[ u_\eta \sin^{2n} 2\phi (2a\Omega \sin \phi - u_\eta \sin^{2n} 2\phi \tan \phi) \right] d\phi + \Phi_0(\eta), \tag{A9}$$

which is solved by dividing the integral into two parts that are solved separately

$$\int u_\eta \sin^{2n} 2\phi \, 2a\Omega \sin \phi d\phi = u_\eta 2a\Omega \int \sin^{2n} 2\phi \sin \phi d\phi \tag{A10a}$$

$$\int -u_\eta^2 \sin^{2n} 2\phi \sin^{2n} 2\phi \tan \phi d\pi = -u_\eta^2 \int \sin^{4n} 2\phi \tan \phi d\phi. \tag{A10b}$$

Eq. (A10a) is solved

$$\int \sin^{2n}(2\phi) \sin \phi d\phi \overset{*}{=} \int (2 \sin \phi \cos \phi)^{2n} \sin \phi d\phi$$

$$= 2^{2n} \int \sin^{2n} \phi \cos^{2n} \phi \sin \phi d\phi$$

$$\overset{**}{=} 4^n \int (1 - \cos^2 \phi)^n \cos^{2n} \phi \sin \phi d\phi,$$

which is integrated by substituting $u = \cos\phi$ and $du = -\sin\phi$

$$4^n \int (1 - \cos^2\phi)^n \cos^{2n}\phi \sin\phi d\phi = -4^n \int (1-u^2)^n u^{2n} du$$

$$\overset{***}{=} -4^n \int \left( \sum_{k=0}^n \binom{n}{k}(-1)^k u^{2(k+n)} \right) du$$

$$= -4^n \sum_{k=0}^n \binom{n}{k}(-1)^k \int u^{2(k+n)} du$$

$$= -4^n \sum_{k=0}^n \binom{n}{k}(-1)^k \frac{1}{2(k+n)+1} u^{2(k+n)+1}$$

$$= -4^n \sum_{k=0}^n \binom{n}{k}(-1)^k \frac{1}{2(k+n)+1} \cos^{2(k+n)+1}\phi.$$

Hence,

$$\int u_\eta \sin^{2n} 2\phi 2a\Omega \sin\phi d\phi = -u_\eta 2a\Omega 4^n \sum_{k=0}^n \binom{n}{k}(-1)^k \frac{1}{2(k+n)+1} \cos^{2(k+n)+1}\phi. \tag{A11}$$

Then, Eq. (A10b) is solved

$$\int \sin^{4n}(2\phi)\tan\phi d\phi \overset{+,*}{=} \int (2\sin\phi\cos\phi)^{4n} \frac{\sin\phi}{\cos\phi} d\phi$$

$$= 2^{4n} \int \sin^{4n+1}\phi \cos^{4n-1}\phi d\phi$$

$$= 16^n \int \sin^{4n+1}\phi \cos^{2(2n-1)}\phi) \cos\phi d\phi$$

$$\overset{**}{=} 16^n \int \sin^{4n+1}\phi(1-\sin^2\phi)^{2n-1}\cos\phi d\phi,$$

which is integrated by substituting $u = \sin\phi$ and $du = \cos\phi$. That gives us

$$16^n \int \sin^{4n+1}\phi(1-\sin^2\phi)^{2n-1}\cos\phi d\phi = 16^n \int u^{4n+1}(1-u^2)^{2n-1} du$$

$$\overset{***}{=} 16^n \int \left( u^{4n+1} \sum_{k=0}^{2n-1} \binom{2n-1}{k}(-1)^k u^{2k} \right) du$$

$$= 16^n \int \left( \sum_{k=0}^{2n-1} \binom{2n-1}{k}(-1)^k u^{2k+4n+1} \right) du$$

$$= 16^n \sum_{k=0}^{2n-1} \binom{2n-1}{k}(-1)^k \int \left( u^{2k+4n+1} \right) du$$

$$= 16^n \sum_{k=0}^{2n-1} \binom{2n-1}{k}(-1)^k \frac{1}{2(k+2n+1)} u^{2(k+2n+1)}$$

$$= 16^n \sum_{k=0}^{2n-1} \binom{2n-1}{k}(-1)^k \frac{1}{2(k+2n+1)} \sin^{2(k+2n+1)} 2\phi.$$

Hence,

$$-u_\eta^2 \int \sin^{4n} 2\phi \tan\phi d\phi = -u_\eta^2 16^n \sum_{k=0}^{2n-1} \binom{2n-1}{k}(-1)^k \frac{1}{2(k+2n+1)} \sin^{2(k+2n+1)} 2\phi. \tag{A12}$$

Combining the solutions of Eqs. (A10a) and (A10b) gives us the solution for $\Phi'$

$$\Phi'(\lambda,\phi,\eta) = -u_\eta 2a\Omega 4^n \sum_{k=0}^{n} \binom{n}{k}(-1)^k \frac{1}{2(k+n)+1} \cos^{2(k+n)+1}\phi \tag{A13}$$

$$-u_\eta^2 16^n \sum_{k=0}^{2n-1} \binom{2n-1}{k}(-1)^k \frac{1}{2(k+2n+1)} \sin^{2(k+2n+1)} 2\phi$$

$$+\Phi_0(\eta).$$

Since the deviations of $\Phi'$ vanishes when averaging horizontally, we solve for $\Phi_0$ by solving the equation

$$\frac{1}{4\pi} \int\limits_{0}^{2\pi} \int\limits_{-\pi/2}^{\pi/2} \Phi'(\lambda,\phi,\eta)\cos\phi d\phi d\lambda = 0 \tag{A14}$$

by inserting the expression for $\Phi'(\lambda,\phi,\eta)$

$$\frac{1}{4\pi} \int\limits_{0}^{2\pi} \int\limits_{-\pi/2}^{\pi/2} [-u_\eta 2a\Omega 4^n \sum_{k=0}^{n} \binom{n}{k}(-1)^k \frac{1}{2(k+n)+1} \cos^{2(k+n)+1}\phi \tag{A15}$$

$$-u_\eta^2 16^n \sum_{k=0}^{2n-1} \binom{2n-1}{k}(-1)^k \frac{1}{2(k+2n+1)} \sin^{2(k+2n+1)} 2\phi$$

$$+\Phi_0(\eta)]\cos\phi d\phi d\lambda = 0.$$

Equation (A15) is divided into three parts

$$\frac{1}{4\pi} \int\limits_{0}^{2\pi} \int\limits_{-\pi/2}^{\pi/2} \left(-u_\eta 2a\Omega 4^n \sum_{k=0}^{n} \binom{n}{k}(-1)^k \frac{1}{2(k+n)+1} \cos^{2(k+n)+1}\phi\right)\cos\phi d\phi d\lambda \tag{A16a}$$

$$\frac{1}{4\pi} \int\limits_{0}^{2\pi} \int\limits_{-\pi/2}^{\pi/2} \left(-u_\eta^2 16^n \sum_{k=0}^{2n-1} \binom{2n-1}{k}(-1)^k \frac{1}{2(k+2n+1)} \sin^{2(k+2n+1)}\phi\right)\cos\phi d\phi d\lambda \tag{A16b}$$

$$\frac{1}{4\pi} \int\limits_{0}^{2\pi} \int\limits_{-\pi/2}^{\pi/2} \Phi_0(\eta)\cos\phi d\phi d\lambda. \tag{A16c}$$

The first part, Eq. (A16a), is solved as follows

$$\frac{1}{4\pi}\int\limits_{0}^{2\pi}\int\limits_{-\pi/2}^{\pi/2}\left(-u_\eta 2a\Omega 4^n\sum_{k=0}^{n}\binom{n}{k}(-1)^k\frac{1}{2(k+n)+1}\cos^{2(k+n)+1}\phi\right)\cos\phi d\phi d\lambda$$

$$=-u_\eta 2a\Omega 4^n\frac{1}{4\pi}\int\limits_{0}^{2\pi}\int\limits_{-\pi/2}^{\pi/2}\left(\sum_{k=0}^{n}\binom{n}{k}(-1)^k\frac{1}{2(k+n)+1}\cos^{2(k+n+1)}\phi\right)d\phi d\lambda$$

$$=-u_\eta 2a\Omega 4^n\frac{1}{4\pi}\int\limits_{0}^{2\pi}\left(\sum_{k=0}^{n}\binom{n}{k}(-1)^k\frac{1}{2(k+n)+1}\int\limits_{-\pi/2}^{\pi/2}\cos^{2(k+n+1)}\phi d\phi\right)d\lambda$$

$$\overset{\dagger}{=}-u_\eta 2a\Omega 4^n\frac{1}{4\pi}\int\limits_{0}^{2\pi}\left(\sum_{k=0}^{n}\binom{n}{k}(-1)^k\frac{1}{2(k+n)+1}\sqrt{\pi}\frac{\Gamma(k+n+3/2)}{\Gamma(k+n+2)}\right)d\lambda$$

$$=-u_\eta 2a\Omega 4^n\frac{1}{4\pi}2\pi\left(\sum_{k=0}^{n}\binom{n}{k}(-1)^k\frac{1}{2(k+n)+1}\sqrt{\pi}\frac{\Gamma(k+n+3/2)}{\Gamma(k+n+2)}\right)$$

$$=-u_\eta a\Omega 4^n\left(\sum_{k=0}^{n}\binom{n}{k}(-1)^k\frac{1}{2(k+n)+1}\sqrt{\pi}\frac{\Gamma(k+n+3/2)}{\Gamma(k+n+2)}\right),$$

where $\Gamma(z)$ is the Gamma function. The second part, Eq. (A16b), is integrated by substituting $u=\sin\phi$ and $du=\cos\phi$ as follows

$$\frac{1}{4\pi}\int\limits_{0}^{2\pi}\int\limits_{-\pi/2}^{\pi/2}\left(u_\eta^2 16^n\sum_{k=0}^{2n-1}\binom{2n-1}{k}(-1)^k\frac{1}{2(k+2n+1)}\sin^{2(k+2n+1)}\phi\right)\cos\phi d\phi d\lambda$$

$$=-u_\eta^2 16^n\frac{1}{4\pi}\int\limits_{0}^{2\pi}\left(\sum_{k=0}^{2n-1}\binom{2n-1}{k}(-1)^k\frac{1}{2(k+2n+1)}\int\limits_{-\pi/2}^{\pi/2}\sin^{2(k+2n+1)}\phi\cos\phi d\phi\right)d\lambda$$

$$\overset{\ddagger}{=}-u_\eta^2 16^n\frac{1}{4\pi}\int\limits_{0}^{2\pi}\left(\sum_{k=0}^{2n-1}\binom{2n-1}{k}(-1)^k\frac{1}{2(k+2n+1)}\frac{2}{2(2n+k+1)+1}\right)d\lambda$$

$$=-u_\eta^2 16^n\frac{1}{4\pi}2\pi\sum_{k=0}^{2n-1}\binom{2n-1}{k}(-1)^k\frac{1}{2(k+2n+1)}\frac{2}{2(2n+k+1)+1}$$

$$=-u_\eta^2\frac{16^n}{2}\sum_{k=0}^{2n-1}\binom{2n-1}{k}(-1)^k\frac{1}{2(2n+k+1)}\frac{2}{2(2n+k+1)+1}$$

    The third part, Eq. (A16c), is solved as

$$\frac{1}{4\pi}\int\limits_{0}^{2\pi}\int\limits_{-\pi/2}^{\pi/2}\Phi_0(\eta)\cos\phi d\phi d\lambda=\Phi_0(\eta),$$

which gives us that

$$\Phi_0(\eta) = -\big((A16a) + (A16b)\big)$$

$$= -\left( -u_\eta a\Omega 4^n \sum_{k=0}^{n} \binom{n}{k}(-1)^k \frac{1}{2(k+n)+1}\sqrt{\pi}\frac{\Gamma(k+n+3/2)}{\Gamma(k+n+2)} \right.$$

$$\left. -u_\eta^2 \frac{16^n}{2} \sum_{k=0}^{2n-1} \binom{2n-1}{k}(-1)^k \frac{1}{2(2n+k+1)}\frac{2}{2(2n+k+1)+1} \right)$$

$$= u_\eta a\Omega 4^n \sum_{k=0}^{n} \binom{n}{k}(-1)^k \frac{1}{2(k+n)+1}\sqrt{\pi}\frac{\Gamma(k+n+3/2)}{\Gamma(k+n+2)}$$

$$+ u_\eta^2 \frac{16^n}{2} \sum_{k=0}^{2n-1} \binom{2n-1}{k}(-1)^k \frac{1}{2(2n+k+1)}\frac{2}{2(2n+k+1)+1}. \tag{A17}$$

Combining Eqs. (A13) and (A17) leads us to the final expression for the deviation of the geopotential field

$$\Phi'(\lambda,\phi,\eta) = u_\eta a\Omega 4^n \left( \underbrace{\sum_{k=0}^{n} \binom{n}{k}(-1)^k \frac{1}{2(k+n)+1}\sqrt{\pi}\frac{\Gamma(k+n+3/2)}{\Gamma(k+n+2)}}_{:=F_3} \right.$$

$$\left. -2\underbrace{\sum_{k=0}^{n} \binom{n}{k}(-1)^k \frac{1}{2(k+n)+1}\cos^{2(k+n)+1}\phi}_{:=F_1} \right)$$

$$+ u_\eta^2 16^n \left( \frac{1}{2}\underbrace{\sum_{k=0}^{2n-1} \binom{2n-1}{k}(-1)^k \frac{1}{2(2n+k+1)}\frac{2}{2(2n+k+1)+1}}_{:=F_4} \right.$$

$$\left. -\underbrace{\sum_{k=0}^{2n-1} \binom{2n-1}{k}(-1)^k \frac{1}{2(k+2n+1)}\sin^{2(k+2n+1)}2\phi}_{:=F_2} \right).$$

It can be re-written as

$$\Phi'(\lambda,\phi,\eta) = u_\eta a\Omega 4^n \left( F_3 - 2F_1 \right) + u_\eta^2 16^n \left( \frac{1}{2}F_4 - F_2 \right), \tag{A18}$$

where

$$F_1 = \sum_{k=0}^{n} \binom{n}{k} (-1)^k \frac{1}{2(k+n)+1} \cos^{2(k+n)+1} \phi \tag{A19a}$$

$$F_2 = \sum_{k=0}^{2n-1} \binom{2n-1}{k} (-1)^k \frac{1}{2(k+2n+1)} \sin^{2(k+2n+1)} 2\phi \tag{A19b}$$

$$F_3 = \sum_{k=0}^{n} \binom{n}{k} (-1)^k \frac{1}{2(k+n)+1} \sqrt{\pi} \frac{\Gamma(k+n+3/2)}{\Gamma(k+n+2)} \tag{A19c}$$

$$F_4 = \sum_{k=0}^{2n-1} \binom{2n-1}{k} (-1)^k \frac{1}{2(2n+k+1)} \frac{2}{2(2n+k+1)+1} \tag{A19d}$$

$$u_\eta = u_0 \ln \eta \exp(-[\ln \eta / b]^2). \tag{A19e}$$

The geopotential field is now described as

$$\Phi(\lambda, \phi, \eta) = \langle \Phi(\eta) \rangle + \Phi'(\lambda, \phi, \eta) \tag{A20}$$

$$= \frac{T_{v,0} g}{\gamma} (1 - \eta^{\frac{R_d \gamma}{g}}) + \Phi'(\lambda, \phi, \eta), \tag{A21}$$

where the expression for $\langle \Phi(\eta) \rangle$ is the expression derived in Sec. A1 and the same as (Eq. (7), Ullrich et al., 2015).

## A3 Virtual Temperature Field

We now have an expression for $\Phi'(\lambda, \phi, \eta)$ and we continue by deriving the virtual temperature anomaly field $T'$ by starting from the hydrostatic balance

$$T_v'(\lambda, \phi, \eta) = -\frac{\eta}{R_d} \frac{\partial \Phi'(\lambda, \phi, \eta)}{\partial \eta}. \tag{A22}$$

We start by inserting the expression for $\Phi'$ Eq. (A18) and $u_\eta$ into Eq. (A22) and then take the derivative with respect to $\eta$

$$T_v'(\lambda, \phi, \eta) = -\frac{\eta}{R_d} \frac{\partial}{\partial \eta} \left[ u_\eta \underbrace{a\Omega 4^n (F_3 - 2F_1)}_{A_1} + u_\eta^2 \underbrace{16^n (\frac{1}{2} F_4 - F_2)}_{A_2} \right]$$

$$= -\frac{\eta}{R_d} \left[ \frac{\partial u_\eta}{\partial \eta} A_1 + 2 u_\eta \frac{\partial u_\eta}{\partial \eta} A_2 \right]$$

$$= -\frac{\eta}{R_d} \left[ \frac{\partial u_\eta}{\partial \eta} (A_1 + 2 u_\eta A_2) \right].$$

As the terms $F_1, F_2, F_3, F_4$ do not depend on $\eta$, we only need to calculate

$$\frac{\partial u_\eta}{\partial \eta} = \frac{\partial}{\partial \eta} (u_0 \ln \eta \exp(-[\ln \eta / b]^2)) = u_0 \left[ \frac{1}{\eta} \exp(-[\ln \eta / b]^2) + \ln \eta \exp(-[\ln \eta / b]^2) \frac{-2}{b^2} \ln \eta \frac{1}{\eta} \right]$$

$$= u_0 \frac{1}{\eta} \exp(-[\ln \eta / b]^2) \left( 1 - \frac{2(\ln \eta)^2}{b^2} \right). \tag{A23}$$

Inserting Eq. (A23) into Eq. (A22) gives us that

$$T_v'(\lambda, \phi, \eta) = \frac{u_0}{R_d} \exp(-[\ln \eta / b]^2) \left( \frac{2(\ln \eta)^2}{b^2} - 1 \right) \left[ a\Omega 4^n \left( F_3 - 2F_1 \right) + 16^n u_\eta \left( F_4 - 2F_2 \right) \right]. \tag{A24}$$

The virtual temperature field is now described by

$$T_v(\lambda, \phi, \eta) = \langle T_v(\eta) \rangle + T_v'(\lambda, \phi, \eta) \tag{A25}$$

$$= T_{v,0} \eta^{\frac{R_d \gamma}{g}} + \frac{u_0}{R_d} \exp(-[\ln \eta / b]^2) \left( \frac{2(\ln \eta)^2}{b^2} - 1 \right) \left[ a\Omega 4^n \left( F_3 - 2F_1 \right) + 16^n u_\eta \left( F_4 - 2F_2 \right) \right], \tag{A26}$$

where $\langle T_v(\eta) \rangle$ is the moist version of (Eq. (10), Ullrich et al., 2015).

*Author contributions.* CB and VS designed the experiments. CB developed the initial background state on OpenIFS and setup the experiments. CB and DvdB validated the initial background states. ME derived the analytical solution. VS supervised the experimentation and production of the manuscript. All authors analysed the results and contributed to the manuscript.

*Competing interests.* The contact author has declared that none of the authors has any competing interests.

*Acknowledgements.* The authors wish to acknowledge CSC – IT Center for Science, Finland, for computational resources. The authors want to thank ECMWF for making OpenIFS available to the University of Helsinki. We thank Kevin Hodges for providing the cyclone tracking code TRACK, Daniel Köhler for the code for the computation of RMSE, Johannes Mikkola for the code for the computation of the skewT diagram presented in the Supplementary material, Jouni Räisänen for useful discussions about the moist primitive equations. We also want to thank Peter Bechtold and Gabriella Szépszó for their help to setup moist baroclinic lifecycle experiment on OpenIFS. This research was supported by the Academy of Finland (grant no 338615). ME also thanks funding received from the European Union's Horizon 2020 research and innovation program under grant agreement No 101003470, the NextGEMS project.

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
