# Peer review of "Analytical and adaptable initial conditions for dry and moist baroclinic waves in the global hydrostatic model OpenIFS (CY43R3)"

_EGUsphere, 2023_

## Referee Comment (RC2)

**Review of the GMDD / EGUsphere Manuscript** https://doi.org/10.5194/egusphere-2023-1078

**Title:** Analytical and adaptable initial conditions for moist baroclinic waves in a global hydrostatic model

**Authors:** Clement Bouvier, Daan van den Broek, Madeleine Ekblom, and Victoria A. Sinclair

**Summary:**
The manuscript describes a new family of analytical initial conditions for atmospheric General Circulation Models which are suitable for the simulation of idealized baroclinic waves on the surface of the sphere. This research is inspired by existing descriptions of baroclinic wave test cases for the dynamical cores of GCMs, and thereby further extends this line of research. The suggested initial conditions consist of a well-balanced background state with two midlatitudinal zonal jets for either dry or moist model configurations. This balanced initial state can either be used as is as a steady-state test case for a dynamical core or can be overlaid with a midlatitudinal zonal wind perturbation to trigger the generation of a baroclinic wave in the midlatitudes. Various options are suggested for the background state which are determined by the chosen parameters for the width of the zonal jets and their vertical center position. These choices determine the baroclinicity and stability characteristics of the initial conditions which then impacts the growth rates and propagation speeds of the baroclinic waves in case a perturbation is chosen.
The newly-derived analytical initial conditions for the moist configuration are then tested in the ECMWF model OpenIFS to demonstrate the characteristics of the baroclinic waves and their sensitivity to the chosen parameters.

**Major comments:**
The research is very interesting and, as mentioned above, extends the already available suite of idealized GCM test configurations with respect to baroclinic wave investigations in spherical geometry. However, there are some major aspects that need attention before a publication can be recommended. They are related to the reproducibility of the results, an error in the formulation of the initial moisture field, and the actual implementation of the initial conditions in OpenIFS. The major concerns are:

1) A particular deficiency is that the presented OpenIFS implementation results are irreproducible by other GCMs since the authors decided to show the simulation results for their moist configuration. The latter utilizes a selected suite of OpenIFS physics parameterizations which are not available in other models. Despite the authors' choice of the moist configuration no attention is paid to the actual impact of the moisture on the simulation which could have served as an interesting talking point. For example, no precipitation or cloud patterns are discussed in the manuscript that would take advantage of the moist configuration and physical parameterizations. Therefore, the question arises why the irreproducible (by other models) moist version was picked here.
A revised version of the manuscript should push the majority of the moist results into a new section, the Appendix, and/or the Supplemental material, and focus the proof-of-concept and sensitivity study for the parameters *n* and *b* on the dry configuration. This way, new users of the test configuration can directly compare their dry results to this manuscript, thereby gaining

confidence that their implementation is correct (provided OpenIFS is correct, see point 4). The title should be more inclusive and state … 'for dry and moist' … instead of only 'moist'.

2) In order to make the results reproducible, important pieces of information about the OpenIFS diffusion settings are required. A listing of the diffusion coefficient, decentering parameter (if used), and the Asselin filter coefficient is needed as the growth rates of the modeled baroclinic waves are impacted by these dissipation choices. For example, is the 4[th]-order horizontal diffusion used as described in Eq. (2.60) in https://www.ecmwf.int/en/elibrary/80319-ifs-documentation-cy43r3-part-iii-dynamics-and-numerical-procedures, e.g. with the specified coefficient? Quote the value for the TL319 resolution.

3) As mentioned in 1) it is left open what the differences between the dry and moist simulations are. A short paragraph/section on the dry/moist differences is desirable. Experiences from the Dynamical Core Model Intercomparison Project (DCMIP) in 2012 and 2016 showed that the presence of idealized precipitation processes intensifies the development of the baroclinic waves in comparison to their dry counterparts. Is this the case here? I recommend adding a time series plot of the minimum mean sea level pressure for both the dry and moist configurations that can display the various growth rates.

4) It would have been beneficial to also see the simulation results for a second non-OpenIFS (dry) dynamical core to gain confidence that the implementation is correct and that two models converge towards a reference solution. This is not a must for the revised version though. However, the relatively slow growth rates for the current moist implementation (and the expected even slower growth rates for a dry implementation) are surprising. The slow growth rates in comparison to other baroclinic wave examples from the literature might be a product of a reduced baroclinicity in this configuration, but this also raises the question whether the OpenIFS configuration works flawlessly.

5) The Zenodo archive distributes the source code for the initial conditions as OpenIFS Fortran code. This means that the initial conditions are not a standalone subroutine that others could just grab and embed into their models. In case the authors would like to promote a wide adoption of the initial conditions by others, they should consider also providing a generic non-OpenIFS version of the initial condition routine.

6) The definition of the saturation vapor pressure (Eq. 13a) is incorrect. This equation (this is the approximation by Bolton (1980)) needs to use the temperature instead of the virtual temperature. All moist simulations will need to be revised after the correction.

7) The authors never define the value for the surface pressure $p_s$, but from the Figs. 5 and 6 as well as the hard-coded value 100000 in the (Zenodo) Fortan code for the moisture initialization, it seems that $p_s = 1000$ hPa is intended and was used for the implementation of the initial conditions in OpenIFS. The information about $p_s$ needs to be provided. Unfortunately, the choice of $p_s = 1000$ hPa leads to an inconsistency between the OpenIFS hybrid vertical coordinate $\eta$ design and the normalized pressure variable defined for the test case $\eta = p/p_s$. This is due to the choice of the reference pressure $p_0 = 1013.25$ hPa in OpenIFS instead of 1000 hPa. OpenIFS defines the eta coordinate as $\eta = a/p_0 + b$ where $a$ (in Pa) and $b$ (unitless) are the hybrid coefficients of the 137 vertical layers. This means that the pressure in OpenIFS is computed as $p = a + b\,p_s$ which corresponds to $p/p_s = a/p_s + b$.

However, using the normalized pressure for the baroclinic wave from the manuscript $\eta = p/p_s$ and plugging in the OpenIFS definition of $\eta$ we get

$$p = \eta p_s = \left(\frac{a}{p_0} + b\right) p_s = a\frac{p_s}{p_0} + bp_s$$

instead of the OpenIFS definition:

$$p = a + bp_s$$

Since $p_s \neq p_0$ this means that the implementation of the current initial conditions in OpenIFS is slightly imbalanced once the first time step is conducted. This can be remedied by either selecting $p_s = 1013.25$ hPa for this test case or rescaling/redefining the OpenIFS hybrid coefficients *'a'* to correct this inconsistency. The latter might be preferred as $p_0 = 1000$ hPa is a popular choice for other models. I suspect that the wavy behavior shown in Fig. 2 for the steady-state condition might actually be caused by this inconsistency (or at least it contributes). In any case, all simulation results will need to be rerun after the correction.

The OpenIFS hybrid coordinate is described in https://www.ecmwf.int/en/elibrary/80319-ifs-documentation-cy43r3-part-iii-dynamics-and-numerical-procedures (e.g. see Eqs. (3.8) and (3.14), also the list of coefficients on the page https://confluence.ecmwf.int/display/OIFS/4.4+OpenIFS%3A+Vertical+Resolution+and+Configurations)

**Minor comments:**
1) There are many small English grammar mistakes or missing words (like 'the' or others) throughout the manuscript. The authors should work with native speakers or professionals to correct these (too many to list them here).
2) Line 5: the statement that a baroclinic wave can only develop if an unbalanced perturbation is used is strictly speaking incorrect. This is typically only true for models on lat-lon grids. If other grids are used, the grid itself is a perturbation and acts as a (slow) trigger for waves. Please rephrase.
3) Line 43-44: The introduction lacks depth/references when it comes to describing the current suite of baroclinic wave test cases for spherical geometry. I suggest adding:
   https://journals.ametsoc.org/view/journals/mwre/132/11/mwr2788.1.xml
   https://rmets.onlinelibrary.wiley.com/doi/abs/10.1002/qj.2241
   The QJ (2006) version of the NCAR Technical Report:
   https://rmets.onlinelibrary.wiley.com/doi/10.1256/qj.06.12
   There is also a new moist and dry variant of the Ullrich et al. (2014) test case with topography as the trigger of the baroclinic wave instead of an overlaid perturbation:
   Hughes, O. K. and C. Jablonowski (2023), A Mountain-Induced Moist Baroclinic Wave Test Case for the Dynamical Cores of Atmospheric General Circulation Models, EGUSphere and Geosci. Model Dev. Discuss., https://egusphere.copernicus.org/preprints/2023/egusphere-2023-376/, in press

   It might also be worth including information about steady-state initial conditions on the sphere like:
   https://rmets.onlinelibrary.wiley.com/doi/abs/10.1002/qj.122

4) Line 59: acronyms Z1, Z2, Z3 and LC1, LC2 and LC3 need some context/explanations
5) Line 75: misleading wording, v is not constant over time, just state that the initial v is set to 0
6) Line 97: $p_s$ is used but never defined, correct, also make sure to state that the topography (surface geopotential) is zero for this initial data set.
7) Line 103: define $R_d$ as the gas constant for dry air
8) Line 161: quote the units for RH (percent). This is also true for Eq. (14): 100 needs to be 100%. I saw in the Fortran implementation that RH is handled as a fraction (between 0-1), therefore the units avoid any confusion here.
9) Line 166: State that Eq. (13a) is the Bolton (1980) approximation. The use of $T_v$ is incorrect. Here, T needs to be used. Specify the units of the numbers in the Bolton equation.
10) Fig.1 is incorrect. There is a direct linear correspondence between $n = p/p_s$ (left axis) and pressure (right axis). The current graph uses a logarithmic relationship which is incorrect. Maybe the authors wanted to show the height along the left axis? Revise the figure.
11) Line 176, revise: negative temperatures in Celsius are allowed.
12) Line 214-215: The original OpenIFS implementation contains the dry and moist variant of the Jablonowski and Williamson (2006) baroclinic wave. The moist variant was used during DCMIP 2012 event (described in
The DCMIP-2012 test case document: Ullrich, P. A., C. Jablonowski, J. Kent, P. H. Lauritzen, R. Nair, M. A. Taylor (2012): Dynamical Core Model Intercomparison Project (DCMIP) Test Case Document, Technical Report, version 1.7 from Jan/13/2013).
The current OpenIFS implementation seems to overwrite the original implementation with the test case numbers 41 and 42, thereby reusing the existing test case infrastructure. Is this correct? I recommend mentioning this.
13) Line 220: What is the relevance of the -1.8C freezing point for water here? The authors define SSTs that are about -20C or lower, therefore it is unclear why -1.8C is emphasized. It is not the actual value of the SST that matters for the surface fluxes and stability, but the jump between the conditions at the surface and the lowest atmospheric layer.
14) Line 231: do not use 'complex' since it alludes to complex number theory (which is not used).
15) Line 236-243: factorials are not actually removed, they are just hidden in the binomials now, revise line 232
16) Lines 244-253 and Table 1 are OpenIFS-specific and better suited as an Appendix with specific OpenIFS implementation details. Consider moving this information. 'namespace' is not the correct phrase, it is called 'namelist'. Line 252 states N3DINI=3, but Table 1 lists N3DINI=2 (contradiction), correct
17) Line 259, use: 'Gaussian hill zonal wind perturbation'
18) Table 1: why is LAPE and LAQUA are set to true in the dry case (as shown in the Zenodo archive), explain the meaning of all namelist settings, the acronyms are too OpenIFS specific to be understood as is by the general audience
19) Table 2: Remove 'Maximum', $u_0$ is not the actual maximum, correct the units of the lapse rate (K/m), use 'Amplitude of the zonal wind pertubation'
20) Provide more insight into the actual resolution. It is stated that TL319 is used which should correspond to a linear grid with 320x640 grid points (in case of the full grid) with a grid spacing of about 62 km. Is the reduced Gaussian grid (N320) or the full grid (F320) used?

However, the Zenodo fort.4 files list the input values
&NAMFPD
NLAT=640,
NLON=1280,
which do not correspond to the N320 but N640 (31 km grid spacing). Please clarify what the actual resolution for the simulations was.

21) Line 270: Provide a reference for L137 level setup (e.g. online page), list the position of the model top

22) Line 279: The symbol w is not the location of the cell interfaces, it is the weight at these locations (typically used as the cos(phi) as a weight that takes the convergence of the meridians into account). The OpenIFS w is the 'Gaussian' weight.

23) Lines 191-294: 'geostrophic' is mentioned here. When plotting the stability parameters in Fig. 3 are indeed the geostrophic definitions use, or the generic ones (without the geostrophic approximation)? Please clarify.

24) Line 310: was a decentering parameter used in OpenIFS? Without decentering, the NCAR CAM SLD T170 RMSE error is only 0.02 m/s at day 15, thereby comparable to OpenIFS. With the decentering activated (the parameter was 0.2) the SLD errors were higher as shown in the referenced NCAR Technical Report.

25) Line 360, use: … vertical temperature gradient in the tropics …

26) Line 363: what is meant by the phrase 'moist T' in the supplement? Is it $T_v$ or T from the moist simulation? Does 'dry T' refer to the temperature in the dry simulation? This needs to be clarified.

27) Line 393: It is not explained whether Fig. 6 shows the moist or dry simulation. I guess it is the moist one. Caption also needs to state this (also true for the Fig. 5 and Fig. 7 captions)

28) Fig.5 and 6: the colors are too dense and hard to distinguish, thin out by a factor of 2 (4C spacing). Does it rain in these simulations? When does the rain start? Comment in the text.

29) Section5: Add (multi-panel) figure to show the time evolution of the minimum surface pressure (for the various n options). No tracker is needed for this. Also expose the evolution of the dry configurations versus the moist one. The growth rates should be different.

30) Line 435: I strongly recommend adding a standalone initialization routine to the Zenodo archive to promote the use of this baroclinic wave configuration

31) Line 449: Explain how $c_p$ is modified when moisture is used.

32) Line 466: correct formatting problem, provide value of $p_s$.

33) Reference: many references are incomplete (see GMD formatting guidelines and correct). Use unique names/acronyms for the journal names, currently it is a mix (e.g. MWR, MON WEATHER REV)

---

## Author Comment (AC2)

Responses to Reviewers' Comments for Manuscript egusphere-2023-1078

**Analytical and adaptable initial conditions for dry and moist baroclinic waves in the global hydrostatic model OpenIFS (CY43R3)**

Addressed Comments for Publication to

Geoscientific Model Development

by

Clément Bouvier, Daan van den Broek, Madeleine Ekblom and Victoria Sinclair Dear Dr. Travis O'Brien,

Please find enclosed the revised version of our previous submission entitled "Analytical and adaptable initial conditions for dry and moist baroclinic waves in the global hydrostatic model OpenIFS (CY43R3)" with manuscript number egusphere-2023-1078. We would like to thank you and the reviewers for the valuable comments which helped improving the quality of our manuscript. In this revision, we have carefully addressed the reviewers' comments. A summary of main modifications and a detailed point-by-point response to the comments from Reviewers 1 and 2 (following the reviewers' order in the decision letter) are given below.

Sincerely,

Clément Bouvier, Daan van den Broek, Madeleine Ekblom and Victoria Sinclair

**Note:** To enhance the legibility of this response letter, all the editor's and reviewers' comments are typeset in boxes. Rephrased or added sentences are typeset in color. The respective parts in the manuscript are highlighted to indicate changes.

**Authors' Response to the Editor**

**General Comments.** In particular, please note that for your paper, the following requirements have not been met in the Discussions paper:

- "The main paper must give the model name and version number (or other unique identifier) in the title."
- 2. "If the model development relates to a single model then the model name and the version number must be included in the title of the paper. If the main intention of an article is to make a general (i.e. model independent) statement about the usefulness of a new development, but the usefulness is shown with the help of one specific model, the model name and version number must be stated in the title. The title could have a form such as, "Title outlining amazing generic advance: a case study with Model XXX (version Y)"."

Therefore please replace the "a global hydrostatic model" in the title of you manuscript by " the global hydrostatic model OpenIFS (CY43R3)".

**Response:** We appreciate your handling of the review process and apologize for the inconsistency with the guidelines.

We changed the title as suggested:

Analytical and adaptable initial conditions for dry and moist baroclinic waves in a global hydrostatic model OpenIFS (CY43R3)

General Comments. Additionally, I want to challenge your statement in the code availability section, that the output can be easily reproduced. Exactly because the OpenIFS model is also not completely open accessible, outputs can not be easily reproduced by every interested reader and as you do not provide your plotting scripts, it can also not be directly be tested if it is infact exactly the same result. Please archive the most important data shown in your paper and provide the plotting scripts for your figures.

**Response:** We want to thank you for this important comment, which will increase the reproductibility of the paper.

We added to the zenodo repository our raw data and cyclone tracked data used to produce the figures and results presented in the preprint. Additionally, we included all plotting scripts used to produce all the figures (https://doi.org/10.5281/zenodo.7890586). We changed the "Code and data availability" section accordingly.

**Authors' Response to Reviewer 1**

**General Comments.** This is a very good manuscript which requires only some very minor edits before publication. It is clear and well written describing analytical formulae for initial conditions for a moist baroclinic instability test case to be used in testing dynamical circulation models. My suggestions for editing are as follows:

**Response:** Thank you for your feedback.

We have carefully addressed all the issues item by item as follows.

Comment 1

Line 5 The statement implies a baroclinic wave will ONLY develop if an unbalanced perturbation is added. This is not so since baroclinic instability is essentially a balanced flow instability, a balanced perturbation can also excite wave development.

**Response:** Thank you for the comment.

We agree with the reviewer's comment and deleted the word "only" in the specified sentence.

**Comment 2**

Line 22 'system' should be 'systems'

**Response:** Thank you for the comment.

We agree with the reviewer's comment and changed the word accordingly.

Line 24 'pattern' should be 'patterns'

**Response:** Thank you for the comment.

We agree with the reviewer's comment and changed the word accordingly.

**Comment 4**

Line 34 'on f - planes' should be 'on an f - plane' and ' $\beta - planes$ ' should be 'a ' $\beta - plane$ '

**Response:** Thank you for the comment.

We agree with the reviewer's comment and changed the words accordingly.

**Comment 5**

Line 36 'used approximation' should be' approximation that is used'

**Response:** Thank you for the comment.

We agree with the reviewer's comment and changed the expression accordingly.

**Comment 6**

Line 94 The zonal wind and gradient and hydrostatic balance, being a balance that is possible in a non-hydrostatic system SHOULD be a valid solution to the full equations of motion, not just the hydrostatic equations.

**Response:** Thank you for the comment. We agree with the reviewer. We changed line 94 from As the geopotential and virtual temperature anomaly fields are derived from hydrostatic equations, the solutions only apply to hydrostatic models.

 $\operatorname{to}$

The geopotential and virtual temperature anomaly fields are derived from hydrostatic equations, and the derived initial states apply to both hydrostatic and non-hydrostatic models.

Comment 7

Equation (16) seems to be missing the terms proportional to  $F_2$  and  $F_4$

**Response:** Thank you for the comment.

At the surface level  $(\eta = 1.0)$ ,  $u_{\eta} = 0.0$  in the Equation (11), which means there is no proportional term to  $F_2$  and  $F_4$  in the Equation (16). To ease the understanding of this fact, we added:

The Sea Surface Temperature (SST) is zonally uniform and is specified to equal the temperature field at  $\eta = 1$  (see Eq. (11) and (15)), which means negative temperatures are allowed and the zonal wind is equal to 0.0 ms-1.

**Comment 8**

Line 243 Since the factorials have been replace by Gamma functions which are DEFINED in terms of factorials, the factorials have not REALLY disappeared .

**Response:** Thank you for the comment.

Effectively, the factorials did not disappear as stated but were combined to Gamma functions, which in turn were combined to obtain binomial coefficients. We corrected this assertion as follow:

In order to avoid the costly use of factorials,  $F_3$  was expressed as a binomial coefficient fraction and all the binomial coefficients were computed once with the multiplicative method, since  $\binom{z}{k+1} = \frac{z-k}{k+1} \binom{z}{k}$  with z and k being integers.

and,

the Gamma function can be replaced by binomial coefficients in  ${\cal F}_3$  as follows

**Comment 9**

Line 356 'increase' should be 'increases'

**Response:** Thank you for the comment.

We agree with the reviewer's comment and changed the word accordingly.

**Comment 10**

Line 421 'use' should be 'use it'

**Response:** Thank you for the comment.

We agree with the reviewer's comment and changed the expression accordingly.

Comment 11

Lines 570 and 571 A factor of  $u_{\eta}$  is missing

**Response:** Thank you for the comment.

Thank you for noting the missing  $u_{\eta}$  on lines 570 and 571. The new version of the manuscript was revised to include the missing  $u_{\eta}$ .

**Authors' Response to Reviewer 2**

General Comments. The research is very interesting and, as mentioned above, extends the already available suite of idealized GCM test configurations with respect to baroclinic wave investigations in spherical geometry. However, there are some major aspects that need attention before a publication can be recommended. They are related to the reproducibility of the results, an error in the formulation of the initial moisture field, and the actual implementation of the initial conditions in OpenIFS.

**Response:** Thank you for your feedback.

We have carefully addressed all the issues item by item as follows.

A particular deficiency is that the presented OpenIFS implementation results are irreproducible by other GCMs since the authors decided to show the simulation results for their moist configuration. The latter utilizes a selected suite of OpenIFS physics parameterizations which are not available in other models. Despite the authors' choice of the moist configuration no attention is paid to the actual impact of the moisture on the simulation which could have served as an interesting talking point. For example, no precipitation or cloud patterns are discussed in the manuscript that would take advantage of the moist configuration and physical parameterizations. Therefore, the question arises why the irreproducible (by other models) moist version was picked here.

A revised version of the manuscript should push the majority of the moist results into a new section, the Appendix, and/or the Supplemental material, and focus the proof-of-concept and sensitivity study for the parameters n and b on the dry configuration. This way, new users of the test configuration can directly compare their dry results to this manuscript, thereby gaining confidence that their implementation is correct (provided OpenIFS is correct, see point 4. The title should be more inclusive and state ... 'for dry and moist' ... instead of only 'moist'.

**Response:** Thank you for the comment.

Our primary motivation for designing these experiments was to allow for a large range of realistic extra-tropical cyclones and hence baroclinic waves, to be simulated so that the dynamics of extra-tropical cyclones in the current and in future climates could be studied. This is why the focus was originally on moist simulations. However this main motivation was not expressed in the introduction because we are aware that this new formation could also be used to test the stability of dynamical cores and we wanted to appeal to both audiences. However, we agree with the reviewer, in that for this new set up to be useful for the comparison and evaluation of dynamical cores, we need to present the results of the dry case with no physics. Therefore, we now include a new sub section showing the evolution of the dry case (Section 5.3.1). Furthermore, as the moist case simulations can be used to study cyclone dynamics, we also expand the results of this section to show the precipitation patterns. Lastly, we changed the last paragraph of the Introduction to explicit better our motivations.

Moreover, the title have been changed to:

Analytical and adaptable initial conditions for dry and moist baroclinic waves in the global hydrostatic model OpenIFS (CY43R3)

**Comment 2**

In order to make the results reproducible, important pieces of information about the OpenIFS diffusion settings are required. A listing of the diffusion coefficient, decentering parameter (if used), and the Asselin filter coefficient is needed as the growth rates of the modeled baroclinic waves are impacted by these dissipation choices. For example, is the 4th-order horizontal diffusion used as described in Eq. (2.60) in OIF [a], e.g. with the specified coefficient? Quote the value for the TL319 resolution.

**Response:** Thank you for the comment.

We agree with the reviewer in that it is important to include all details of the numerical set up, including details on the diffusion. We used the default options in OpenIFS, which are the same as the default options in CY43R3 of the full IFS. In this version there is no decentering nor Asselin filter. The spectral diffusion used by default is of 4th order (with the exponent of the wavenumber dependency REXPDH=4) and is set to be rather weak, the strength of which is related to the used model timestep. The coefficients for  $T_L319$  are 2100.0 seconds (vorticity (HDIRVOR), divergence (HDIRDIV), temperature (HDIRT), humidity (HDIRQ) diffusions) and the other are set to zero. This information

was added in section 3.3.

**Comment 3**

As mentioned in 1 it is left open what the differences between the dry and moist simulations are. A short paragraph/section on the dry/moist differences is desirable. Experiences from the Dynamical Core Model Intercomparison Project (DCMIP) in 2012 and 2016 showed that the presence of idealized precipitation processes intensifies the development of the baroclinic waves in comparison to their dry counterparts. Is this the case here? I recommend adding a time series plot of the minimum mean sea level pressure for both the dry and moist configurations that can display the various growth rates.

**Response:** Thank you for the comment.

This is a good suggestion and in the revised version of the manuscript we include a subsection describing the evolution of the dry case, then the moist case and highlight the main differences. We added Figure 8 showing the precipitation evolution in the moist case.

Moreover, we added Figure 11 comparing the evolution of the 850 hPa vorticities for different n and between the moist and dry cases. As explained in section 5.3, cyclones developing in the dry cases which are run with no physics, reached higher vorticity levels but slower than their moist counterparts which are run with physics. Also note that friction which is also included in our simulations, acts to weaken the cyclones [Boutle et al., 2010].

It would have been beneficial to also see the simulation results for a second non-OpenIFS (dry) dynamical core to gain confidence that the implementation is correct and that two models converge towards a reference solution. This is not a must for the revised version though. However, the relatively slow growth rates for the current moist implementation (and the expected even slower growth rates for a dry implementation) are surprising. The slow growth rates in comparison to other baroclinic wave examples from the literature might be a product of a reduced baroclinicity in this configuration, but this also raises the question whether the OpenIFS configuration works flawlessly.

**Response:** Thank you for the comment.

Unfortunately it is beyond the scope of this study to produce the simulations with another dynamical core as this would entail a huge amount of work.

The slower growth rates of the cyclones in our new case are very likely due to the reduced baroclinicity in our set up compared to the Jablonowski and Williamson [2006] case (a gradient between the poles and equator of 36°C in our case against more than 80°C in the Jablonowski and Williamson [2006] case). One motivation for the development of this new background state was to produce a background state and subsequently cyclones which are much more flexible with more degrees of freedom than in the Jablonowski case Jablonowski and Williamson [2006]. Moreover, the slower growth rates observed in the paper can be explained by the friction induced by the Charnock value (see 3 and section 5.4 of the corrected version).

The Zenodo archive distributes the source code for the initial conditions as OpenIFS Fortran code. This means that the initial conditions are not a standalone subroutine that others could just grab and embed into their models. In case the authors would like to promote a wide adoption of the initial conditions by others, they should consider also providing a generic non-OpenIFS version of the initial condition routine.

**Response:** Thank you for the comment.

We agree with reviewer's comment and included a standalone version in the Zenodo archive. This standalone have been compiled with GFortran version 8.5.0 and tested on Red Hat 8.5.0-10 (https://docs.csc.fi/). This standalone is divided in two parts: (1) a main program setting all the variables to compute the zonal fields and (2) a subroutine computing the zonal fields detailed in sections 2.1 and 2.2 of the manuscript.

**Comment 6**

The definition of the saturation vapor pressure (Eq. 13a) is incorrect. This equation (this is the approximation by Bolton (1980)) needs to use the temperature instead of the virtual temperature. All moist simulations will need to be revised after the correction.

**Response:** Thank you for the comment.

Yes, equation 13a in the manuscript is incorrect and  $T_v$  should be T in this equation. This has now been corrected and we now also state in the revised manuscript that this equation is the Bolton approximation.

The specific humidity field  $q(\lambda, \phi, \eta)$  is then computed to ensure concordance with the proposed virtual temperature and jet structure by assuming  $T = T_v$ . The specific humidity field is derived from the relative humidity  $(RH(\eta))$ , the saturation vapour pressure  $(e_s)$  and the saturation mixing ratio  $(w_s)$  using the Bolton approximation for the saturation vapour pressure [Bolton, 1980, Yau and Rogers, 1996] as presented in the following equations

$$e_s(\lambda, \phi, \eta) = 611.21 \exp \frac{17.67(T(\lambda, \phi, \eta) - 273.15)}{T(\lambda, \phi, \eta) - 29.65}$$
 and (1a)

$$w_s(\lambda, \phi, \eta) = 0.622 \frac{e_s(\lambda, \phi, \eta)}{p(\lambda, \phi, \eta) - e_s(\lambda, \phi, \eta)},$$
(1b)

where  $T(\lambda, \phi, \eta)$  is the temperature field (K),  $e_s(\lambda, \phi, \eta)$  is the saturation vapour pressure (Pa) and p is the pressure (Pa).

With only  $T_v$  and RH it is not possible to compute T and hence q. Therefore we needed to implement an iterative solution to obtain T (see Figure 2 in the revised manuscript). To obtain the final T, we implemented an iterative scheme. On the first iteration, we assume that  $T = T_v$  in the Bolton approximation. Then, the estimated T is used again in the Bolton approximation to estimate a new T. This cycle is repeated 10 times to obtain the final T and q.

The authors never define the value for the surface pressure ps, but from the Figs. 5 and 6 as well as the hard-coded value 100000 in the (Zenodo) Fortan code for the moisture initialization, it seems that  $p_s = 1000hPa$  is intended and was used for the implementation of the initial conditions in OpenIFS. The information about  $p_s$  needs to be provided. Unfortunately, the choice of  $p_s = 1000hPa$  leads to an inconsistency between the OpenIFS hybrid vertical coordinate h design and the normalized pressure variable defined for the test case  $\eta = p/p_s$ . This is due to the choice of the reference pressure  $p_0 = 1013.25hPa$  in OpenIFS instead of 1000hPa. OpenIFS defines the eta coordinate as  $\eta = a/p_0 + b$  where a (in Pa) and b (unitless) are the hybrid coefficients of the 137 vertical layers. This means that the pressure in OpenIFS is computed as  $p = a + bp_s$  which corresponds to  $p/p_s = a/p_s + b$ . However, using the normalized pressure for the baroclinic wave from the manuscript  $\eta = p/p_s$  and plugging in the OpenIFS definition of  $\eta$  we get:

$$p = \eta p_s = (a/p_0 + b)p_s = ap_s/p_0 + bp_s$$
(2)

instead of the OpenIFS definition:

$$p = a + bp_s \tag{3}$$

Since  $p_s \neq p_0$  this means that the implementation of the current initial conditions in OpenIFS is slightly imbalanced once the first time step is conducted. This can be remedied by either selecting  $p_s = 1013.25hPa$  for this test case or rescaling/redefining the OpenIFS hybrid coefficients 'a' to correct this inconsistency. The latter might be preferred as  $p_0 = 1000hPa$  is a popular choice for other models. I suspect that the wavy behavior shown in Fig. 2 for the steady-state condition might actually be caused by this inconsistency (or at least it contributes). In any case, all simulation results will need to be rerun after the correction.

The OpenIFS hybrid coordinate is described in OIF [a] (e.g. see Eqs. (3.8) and (3.14), also the list of coefficients on the page OIF [b])

**Response:** Thank you for the comment.

We did set the surface pressure to 1000 hPa in these experiments, which as the reviewer states, is not fully consistent with the definition of the vertical coordinate. We have re-run the simulations with a surface pressure of 1013.25 hPa and changed the text accordingly. As predicted by the reviewer, the RMSE computed from the new simulations with the corrected surface pressure (the revised Figure 2, now Figure 3) does not present any wavy pattern.

**Comment 8**

There are many small English grammar mistakes or missing words (like 'the' or others) throughout the manuscript. The authors should work with native speakers or professionals to correct these (too many to list them here).

**Response:** Thank you for the comment.

We have carefully proofread the revised manuscript and have hopefully now corrected these minor issues.

**Comment 9**

Line 5: the statement that a baroclinic wave can only develop if an unbalanced perturbation is used is strictly speaking incorrect. This is typically only true for models on lat-lon grids. If other grids are used, the grid itself is a perturbation and acts as a (slow) trigger for waves. Please rephrase.

**Response:** Thank you for the comment.

We agree with the reviewer's comment and deleted the word "only" in the specified sentence.

Line 43-44: The introduction lacks depth/references when it comes to describing the current suite of baroclinic wave test cases for spherical geometry. I suggest adding:

https://journals.ametsoc.org/view/journals/mwre/132/11/mwr2788.1.
xml

https://rmets.onlinelibrary.wiley.com/doi/abs/10.1002/qj.2241

The QJ (2006) version of the NCAR Technical Report:

https://rmets.onlinelibrary.wiley.com/doi/10.1256/qj.06.12

There is also a new moist and dry variant of the Ullrich et al. (2014) test case with topography as the trigger of the baroclinic wave instead of an overlaid perturbation: Hughes, O. K. and C. Jablonowski (2023), A Mountain-Induced Moist Baroclinic Wave Test Case for the Dynamical Cores of Atmospheric General Circulation Models, EGUSphere and Geosci. Model Dev. Discuss., https:// egusphere.copernicus.org/preprints/2023/egusphere-2023-376/, in press. It might also be worth including information about steady-state initial conditions on the sphere like:

https://rmets.onlinelibrary.wiley.com/doi/abs/10.1002/qj.122 https://rmets.onlinelibrary.wiley.com/doi/full/10.1002/asl.349

**Response:** Thank you for the comment.

We agree with the reviewer's comment and included the references in the Introduction.

**Comment 11**

Line 59: acronyms Z1, Z2, Z3 and LC1, LC2 and LC3 need some context/explanations

**Response:** Thank you for the comment.

We agree with the reviewer's comment and changed the text accordingly:

Popular zonal jet structures are Zonal jet 1, Zonal jet 2 and Zonal jet 3 (denoted Z1, Z2 and Z3) resulting in, respectively, baroclinic lifecycles 1, 2 and 3 (denoted LC1, LC2 and LC3) [Thorncroft et al., 1993, Agustí-Panareda et al., 2005].

**Comment 12**

Line 75: misleading wording, v is not constant over time, just state that the initial v is set to 0

**Response:** Thank you for the comment.

We agree with the reviewer's comment and changed the sentence as follow:

In other words, the meridional wind speed is set to  $0.0 \text{ ms}^{-1}$  which leads to a gradient-wind balance.

**Comment 13**

Line 97:  $p_s$  is used but never defined, correct, also make sure to state that the topography (surface geopotential) is zero for this initial data set.

**Response:** Thank you for the comment.

We agree with the reviewer that defining  $p_s$  is necessary. We have corrected this sentence:

The vertical levels are pressure levels normalised with respect to the surface pressure, defined as  $\eta = p/p_s$ , where p is the pressure on the model level and  $p_s$  is the surface pressure. In OpenIFS, the vertical  $\eta$  levels are defined as  $\eta = p/p_s = a/p_s + b$ , where  $p_s = 1013.25$  hPa is the pressure at the surface pressure, and a and b are hybrid coefficients defined for each vertical resolution.

We have clarified that no topography is used in this study by adding to line 93:

The derivation of the analytical initial conditions for geopotential and virtual temperature fields starts from the primitive equations for moist adiabatic and frictionless flow in spherical coordinates and normalised pressure levels for a planet with no topography (i.e., surface geopotential is zero).

**Comment 14**

Line 103: define  $R_d$  as the gas constant for dry air

**Response:** Thank you for the comment.

We agree with the reviewer's comment and changed  $R_d$  definition accordingly.

**Comment 15**

Line 161: quote the units for RH (percent). This is also true for Eq. (14): 100 needs to be 100%. I saw in the Fortran implementation that RH is handled as a fraction (between 0-1), therefore the units avoid any confusion here.

**Response:** Thank you for the comment.

We agree with the reviewer's comment and changed Equations (12) and (14) to include the percentage units.

Line 166: State that Eq. (13a) is the Bolton (1980) approximation. The use of  $T_v$  is incorrect. Here, T needs to be used. Specify the units of the numbers in the Bolton equation.

**Response:** Thank you for the comment.**

As stated above in response to comment number 6, we have changed the section to include the units.

The specific humidity field  $q(\lambda, \phi, \eta)$  is then computed to ensure concordance with the proposed virtual temperature and jet structure by assuming  $T = T_v$ . The specific humidity field is derived from the relative humidity  $(RH(\eta))$ , the saturation vapour pressure  $(e_s)$  and the saturation mixing ratio  $(w_s)$  using the Bolton approximation for the saturation vapour pressure [Bolton, 1980, Yau and Rogers, 1996] as presented in the following equations

$$e_s(\lambda, \phi, \eta) = 611.21 \exp \frac{17.67(T(\lambda, \phi, \eta) - 273.15)}{T(\lambda, \phi, \eta) - 29.65}$$
 and (4a)

$$w_s(\lambda,\phi,\eta) = 0.622 \frac{e_s(\lambda,\phi,\eta)}{p(\lambda,\phi,\eta) - e_s(\lambda,\phi,\eta)},$$
(4b)

where  $T(\lambda, \phi, \eta)$  is the temperature field (K),  $e_s(\lambda, \phi, \eta)$  is the saturation vapour pressure (Pa) and p is the pressure (Pa).

**Comment 17**

Fig.1 is incorrect. There is a direct linear correspondence between  $n = p/p_s$  (left axis) and pressure (right axis). The current graph uses a logarithmic relationship which is incorrect. Maybe the authors wanted to show the height along the left axis? Revise the figure.

**Response:** Thank you for the comment.

We have revised Figure 1 and relative humidity is now plotted as a function of  $\eta$  with a linear relationship between the left-hand y-axis ( $\eta$ ) and the right-hand u-axis (p (hPa)). As a result, there is now only one line showing the relative humidity as a function of  $\eta$  and p (with  $p_s = 1013.25$  hPa and  $p_{top} = 0$ ).

Comment 18

Line 176, revise: negative temperatures in Celsius are allowed.

**Response:** Thank you for the comment.

We agree with the reviewer's comment and changed the text accordingly:

The Sea Surface Temperature (SST) is zonally uniform and is specified to equal the temperature field at  $\eta = 1$  (see Eq. (11) and (15)), which means negative temperatures are allowed and the zonal wind is equal to 0.0 ms-1.

**Comment 19**

Line 214-215: The original OpenIFS implementation contains the dry and moist variant of the Jablonowski and Williamson (2006) baroclinic wave. The moist variant was used during DCMIP 2012 event (described in The DCMIP-2012 test case document: Ullrich, P. A., C. Jablonowski, J. Kent, P. H. Lauritzen, R. Nair, M. A. Taylor (2012): Dynamical Core Model Intercomparison Project (DCMIP) Test Case Document Dyn, Technical Report, version 1.7 from Jan/13/2013). The current OpenIFS implementation seems to overwrite the original implementation with the test case numbers 41 and 42, thereby reusing the existing test case infrastructure. Is this correct? I recommend mentioning this.

**Response:** Thank you for the comment.

We agree with the reviewer, the test case have been overwritten to include the our initial state. We mention in section 3.2 that:

Originally, this background state was implemented in the full version of the IFS, and hence OpenIFS, to test the dynamical core. It was attributed the NTESTCASE 41 (dry case) and 42 (moist case) for the Dynamical Core Intercomparison Project (DCMIP).

Moreover, we changed section 3.4 to state:

The dry and moist test case can be computed by setting the NTESTCASE value to 41 or 42 respectively, replacing *de facto* the previous implementation.

**Comment 20**

Line 220: What is the relevance of the -1.8C freezing point for water here? The authors define SSTs that are about -20C or lower, therefore it is unclear why -1.8C is emphasized. It is not the actual value of the SST that matters for the surface fluxes and stability, but the jump between the conditions at the surface and the lowest atmospheric layer.

**Response:** Thank you for the comment.

We agree with the reviewer's comment. -1.8°C is irrelevant for our configuration and we have removed it from the text. Moreover, we clarified the text:

The original initial state of Jablonowski and Williamson [2006] has a very strong meridional temperature gradient which means that the near-surface temperature reaches -50°C at high latitudes. In the dry case with no physics, the surface heat

fluxes are not computed meaning that the SSTs can be specified to be much warmer (or colder) than the near-surface atmospheric temperatures without causing any problems such as destabilisation of the boundary layer or convection. In contrast, in the moist case with physics on an aquaplanet, exceptionally cold conditions at high latitudes with physically realistic SSTs cause large surface heat fluxes to develop and in the extreme case can result in low pressure centres resembling polar lows developing at high latitudes. Therefore, modifications to the Jablonowski and Williamson [2006] case are needed to enable it to be run with physics and to allow it to be used to investigate cyclone dynamics rather than the numerical accuracy of dynamical cores. Hence, the SST definition presented Section 2.3.

**Comment 21**

Line 231: do not use 'complex' since it alludes to complex number theory (which is not used).

**Response:** Thank you for the comment.

We agree with the reviewer and changed the word 'complex' to 'non-trivial'.

**Comment 22**

Line 236-243: factorials are not actually removed, they are just hidden in the binomials now, revise line 232

**Response:** Thank you for the comment.

Effectively, this was not clearly explained in the original version of the manuscript which has caused some confusion. The factorials have not been removed as stated but were combined to Gamma functions, which in turn were combined to obtain binomial coefficients. We corrected this assertion as follow:

In order to avoid the costly use of factorials,  $F_3$  was expressed as a binomial coefficient fraction and all the binomial coefficients were computed once with the multiplicative method, since  $\binom{z}{k+1} = \frac{z-k}{k+1} \binom{z}{k}$  with z, k integers.

and,

the Gamma function can be replaced by binomial coefficients in  $F_3$  as follows

**Comment 23**

Lines 244-253 and Table 1 are OpenIFS-specific and better suited as an Appendix with specific OpenIFS implementation details. Consider moving this information. 'namespace' is not the correct phrase, it is called 'namelist'. Line 252 states N3DINI=3, but Table 1 lists N3DINI=2 (contradiction), correct

**Response:** Thank you for the comment.

Table 1 is updated with an additional column "Explanation" that explains the meaning of the different parameters. We have correct the word 'namespace' to 'namelist' in the caption of Table 1. Thank you for noting the contradiction of N3DINI; it is now corrected to 2 on line 252.

**Comment 24**

Line 259, use: 'Gaussian hill zonal wind perturbation'

**Response:** Thank you for the comment.

We agree with the reviewer's comments and changed the expression accordingly.

**Comment 25**

Table 1: why is LAPE and LAQUA are set to true in the dry case (as shown in the Zenodo archive), explain the meaning of all namelist settings, the acronyms are too OpenIFS specific to be understood as is by the general audience

**Response:** Thank you for the comment.

As stated in 23, we added a new column to disambiguate the parameters presented in Table 1.

**Comment 26**

Table 2: Remove 'Maximum',  $u_0$  is not the actual maximum, correct the units of the lapse rate (K/m), use 'Amplitude of the zonal wind pertubation'

**Response:** Thank you for the comment.

We agree with the reviewer that  $u_0$  is not the maximum of the wind speed. The amplitude of the zonal wind speed is not defined by  $u_0$  alone but also by the parameter b in the expression for  $u(\lambda, \phi, \eta) = -u_0 \ln(\eta) \exp[-(\frac{\ln \eta}{b})^2] \sin^{2n}(2\phi)$ . We changed the explanation for  $u_0$  from 'maximum zonal wind speed' to 'Together with b,  $u_0$  adjusts the amplitude of zonal mean wind speed  $(ms^{-1})$ '.

We corrected the explanation for  $u_p$  to 'Amplitude of the zonal wind perturbation' and the units of the lapse rate to K/m.

Provide more insight into the actual resolution. It is stated that TL319 is used which should correspond to a linear grid with 320x640 grid points (in case of the full grid) with a grid spacing of about 62 km. Is the reduced Gaussian grid (N320) or the full grid (F320) used? However, the Zenodo fort.4 files list the input values: &NAMFPD NLAT=640, NLON=1280, which do not correspond to the N320 but N640 (31 km grid spacing). Please clarify what the actual resolution for the simulations was.

**Response:** Thank you for the comment.**

The model resolution is  $T_L319$  and thus there are 320 by 640 grid points on the full grid. The values in the fort.4 were misleading. This is because in the idealised model set up, OpenIFS does not read these resolution values from the fort.4 files; NLAT and NLON in fort.4 are ignored. Instead, OpenIFS reads the initial condition files and obtains the grid information from them. Note that it is only the grid information from these initialisation files that OpenIFS reads; all other variables in these files are ignored. The inconsistency spotted by the reviewer's comment have been corrected in the fort.4 files in the Zenodo archive.

**Comment 28**

Line 270: Provide a reference for L137 level setup (e.g. online page), list the position of the model top

**Response:** Thank you for the comment.

We agree with the reviewer and added after  $T_L319 L137$

and 137 vertical levels with a model top of 0.01 hPa (https://confluence.ecmwf. int/display/UDOC/L137+model+level+definitions, accessed: 2023-12-05))

**Comment 29**

Line 279: The symbol w is not the location of the cell interfaces, it is the weight at these locations (typically used as the cos(phi) as a weight that takes the convergence of the meridians into account). The OpenIFS w is the 'Gaussian' weight.

**Response:** Thank you for the comment.

We agree with the reviewer's comment and changed the text as follow:

where  $u_{za}$  is the zonal average of the zonal wind speed,  $u_{ideal}$  is the ideal zonal average of the zonal wind speed computed from the analytical expression for the zonal wind (Eq. (3)),  $w_{\phi_j}$  is the weights to correct the convergence of the meridians  $\phi_j$  and  $\Delta \eta_i$  is the thickness of the model layer  $\eta_i$ .

**Comment 30**

Lines 191-294: 'geostrophic' is mentioned here. When plotting the stability parameters in Fig. 3 are indeed the geostrophic definitions use, or the generic ones (without the geostrophic approximation)? Please clarify.

**Response:** Thank you for the comment.

As the reviewer stated, we used the generic absolute vorticity and not the geostrophic approximation. We changed the text accordingly:

For the initial state to be stable to horizontal displacements (inertial stability) the absolute vorticity must be positive (negative) in the northern (southern) hemisphere. Situations can exist where the atmosphere is statically and inertially stable, but the atmosphere is unstable to slantwise displacements (symmetric instability). This exists when the potential vorticity is negative.

**Comment 31**

Line 310: was a decentering parameter used in OpenIFS? Without decentering, the NCAR CAM SLD T170 RMSE error is only 0.02m/s at day 15, thereby comparable to OpenIFS. With the decentering activated (the parameter was 0.2) the SLD errors were higher as shown in the referenced NCAR Technical Report.

**Response:** Thank you for the comment.

We did not use a decentering parameter. Moreover, with the correction of the surface pressure and the temperature field, the obtained RMSE for all studied cases is significantly lower than the NCAR CAM SLD T170 RMSE.

**Comment 32**

Line 360, use: ... vertical temperature gradient in the tropics ...

**Response:** Thank you for the comment.

We agree with the reviewer's comment and changed the expression accordingly.

Line 363: what is meant by the phrase 'moist T' in the supplement? Is it  $T_v$  or T from the moist simulation? Does 'dry T' refer to the temperature in the dry simulation? This needs to be clarified.

**Response:** Thank you for the comment.

We agree with the reviewer's comment and changed the caption of Figure S3 accordingly

Figure S3: Sounding for two cases (n=3 and b=2.0, n=1 and b=1.0) for two latitudes  $(0^{\circ} \text{ and } 45^{\circ}\text{N})$ . The solid green line represents the dew point, the solid red line the real temperature of the moist case, the solid black line the ideal parcel profile from the surface temperature and the dashed red line the temperature of the dry case (added for reference). The red area represents the convective available potential energy and the blue area the convective inhibition.

**Comment 34**

Line 393: It is not explained whether Fig. 6 shows the moist or dry simulation. I guess it is the moist one. Caption also needs to state this (also true for the Fig. 5 and Fig. 7 captions)

**Response:** Thank you for the comment.

We agree with the reviewer's comment and updated the captions accordingly.

Fig.5 and 6: the colors are too dense and hard to distinguish, thin out by a factor of 2 (4C spacing). Does it rain in these simulations? When does the rain start? Comment in the text.

**Response:** Thank you for the comment.**

We agree with the reviewer's comment and we decreased the spacing in the figures as suggested. Additionally, we also thinned out the spacing of MSLP with a factor of 2, to improve the clarity of the figure.

Moreover, we added a figure with the development of the baroclinic wave which also includes the precipitation (Figure 8 in Section 5.3). We added comments in the text of section 5.3 accordingly.

**Comment 36**

Section5: Add (multi-panel) figure to show the time evolution of the minimum surface pressure (for the various n options). No tracker is needed for this. Also expose the evolution of the dry configurations versus the moist one. The growth rates should be different.

**Response:** Thank you for the comment.

As stated above in response to comment number 3, we have added Figure 11 which compares the evolution of the 850-hPa vorticity for different n and both the moist and dry cases. As several baroclinic waves are developing during the simulation (for all values of n), we deemed it important to be able to separate the different waves and study the evolution of the vorticity of the first cyclone. That is the reason why we chose to use a tracking algorithm with the vorticity at 850hPa.

Line 435: I strongly recommend adding a standalone initialization routine to the Zenodo archive to promote the use of this baroclinic wave configuration

**Response:** Thank you for the comment.

We agree with the reviewer's comment. And as stated in 5, a standalone version is included in the Zenodo archive. Moreover, we changed the Code and Data Availability as follow:

The licence for using the OpenIFS model can be requested from ECMWF user support (openifs-support@ecmwf.int). The modified subroutines of OpenIFS, a standalone version to compute the zonal fields detailed in Sections 2.1 and 2.2, the submission and plotting scripts, the configuration files and the raw data are available on Zenodo (https://doi.org/10.5281/zenodo.7890586).

**Comment 38**

Line 449: Explain how  $c_p$  is modified when moisture is used.

**Response:** Thank you for the comment.

Thank you for pointing out the missing explanation on how to modify  $c_p$  when adding moisture. We have changed the sentence from

Please note that the specific heat capacity of air,  $c_p$ , in the thermodynamic equation needs to be corrected with a correction factor when moisture is included in the model. Please note that the specific heat capacity of air,  $c_p$ , in the thermodynamic equation needs to be corrected with a correction factor  $\delta$  when moisture is included in the model. The correction factor is defined as

$$\delta = 1 + (c_{p,vap}/c_{p,dry} - 1)q,$$

where  $c_{p,vap}/c_{p,dry} = 1860/1004$  (units: J/(kg K)) and q is the specific humidity (units: kg/kg) [ECMWF, 2016, eq. (2.3)].

**Comment 39**

Line 466: correct formatting problem, provide value of  $p_s$ .

**Response:** Thank you for the comment.

We want to thank the reviewer for noting the formatting problem; it is now corrected. The value of  $p_s$  is given in the main text (line: 102) and the derivation of the equations in the Appendix do not use any specific values for  $p_s$ .

**Comment 40**

Reference: many references are incomplete (see GMD formatting guidelines and correct). Use unique names/acronyms for the journal names, currently it is a mix (e.g. MWR, MON WEATHER REV)

**Response:** Thank you for the comment.

Thank you for pointing out the inconsistencies regarding the journal acronyms. The acronyms are now corrected according to the guidelines of GMD.

**References**

- Ifs documentation cy43r3 part iii: Dynamics and numerical procedures. https://www.ecmwf.int/en/elibrary/ 80319-ifs-documentation-cy43r3-part-iii-dynamics-and-numerical-procedures, a. Accessed: 2023-11-21.
- IA Boutle, RJ Beare, Stephen Ernest Belcher, AR Brown, and Robert Stephen Plant. The moist boundary layer under a mid-latitude weather system. *Boundary-layer meteorology*, 134:367–386, 2010.
- Christiane Jablonowski and David L Williamson. A baroclinic instability test case for atmospheric model dynamical cores. *QJ*, 132(621C):2943–2975, 2006.
- David Bolton. The computation of equivalent potential temperature. *MWR*, 108(7): 1046–1053, 1980.
- Man Kong Yau and Roddy Rhodes Rogers. A short course in cloud physics. Elsevier, 1996.
- 4.4 openifs: Vertical resolution and configurations. https://confluence.ecmwf.int/ display/OIFS/4.4+OpenIFS%3A+Vertical+Resolution+and+Configurations, b. Accessed: 2023-11-21.
- CD Thorncroft, BJ Hoskins, and ME McIntyre. Two paradigms of baroclinic-wave life-cycle behaviour. *QJ*, 119(509):17–55, 1993.
- Anna Agustí-Panareda, Suzanne L Gray, George C Craig, and Chris Thorncroft. The extratropical transition of tropical cyclone lili (1996) and its crucial contribution to a moderate extratropical development. MWR, 133(6):1562–1573, 2005.
- Dynamical core model intercomparison project (dcmip) test case document. https://admg.engin.umich.edu/wp-content/uploads/sites/525/2021/03/ DCMIP-2012\_TestCaseDocument\_v1.7.pdf. Accessed: 2023-11-21.
- ECMWF. IFS Documentation CY43R1 Part III: Dynamics and Numerical Procedures. Number 3. ECMWF, 2016 2016. doi: 10.21957/m1u2yxwrl. URL https://www.ecmwf. int/node/17116.

---

## Author Response (AR2)

**Analytical and adaptable initial conditions for dry and moist baroclinic waves in the global hydrostatic model OpenIFS (CY43R3)**

Addressed Comments for Publication to

Geoscientific Model Development

by

Clément Bouvier, Daan van den Broek, Madeleine Ekblom and Victoria Sinclair

Dear Dr. Travis O'Brien,

Please find enclosed the final revised version of our previous submission entitled "Analytical and adaptable initial conditions for dry and moist baroclinic waves in the global hydrostatic model OpenIFS (CY43R3)" with manuscript number egusphere-2023-1078. We would like to thank you for the handling of the review process and your relevant comments. In this final revision, we have carefully addressed your comments. A summary of the main modifications and a detailed point-by-point response are given below.

Sincerely,

Clément Bouvier, Daan van den Broek, Madeleine Ekblom and Victoria Sinclair

**Authors' Response to the Editor**

> **General Comments.** Thank you for all the work from you and your co-authors in addressing the reviewer concerns. This is a very cool paper. At this stage I am happy to accept this paper for publication in GMD pending some technical revisions. The revisions mainly involve modifying language to avoid ambiguity, though I do have an apparent discrepancy to note.

**Response:** We appreciate your handling of the review process.

We carefully answered your remarks as follow.

**Comment 1**

l18: suggest rewording to "create a large ensemble of baroclinic lifecycles"

**Response:** Thank you for the comment.

We agree with the editor's comment and changed the sentence accordingly.

**Comment 2**

l22: I suggest rewording this sentence as "These GCMs provide numerical solutions to the governing equations..."

**Response:** Thank you for the comment.

We agree with the editor's comment and changed the sentence accordingly.

**Comment 3**

l57: you use the term "run without physics" which is potentially ambiguous (since GCMs numerically approximate conservation equations that are derived from first-principles physics). I take it that you mean "run without physics parameterizations"? If so, I suggest using that rewording. There are some other places in the text where the term "physics" should be reworded as "physics parameterization(s)".

**Response:** Thank you for the comment.

We agree with the editor's comment and changed the expression as "physics parameterisation scheme(s)" through all the text.

**Comment 4**

l60: I suggest adding "e.g., " before "Kuo et al., 1991,..." since these are only a representative sample of papers on this topic

**Response:** Thank you for the comment.

We agree with the editor's comment and changed the sentence accordingly.

**Comment 5**

l65: "physics" -> "physics parameterizations"

**Response:** Thank you for the comment.

We agree with the editor's comment and changed the expression as explained in 3.

**Comment 6**

l218: The last sentence of this paragraph seems like a non-sequitur. Consider whether it is necessary here.

**Response:** Thank you for the comment.

We agree with the editor's comment and removed the sentence.

**Comment 7**

l230: "Implementation into OpenIFS" -> "Implementation in OpenIFS"

**Response:** Thank you for the comment.

We agree with the editor's comment and changed the title accordingly.

**Comment 8**

l244: I suggest rewording th sentence "It was attributed..." as "This background state was referred to as NTESTCASE41 (dry case)..."

**Response:** Thank you for the comment.

We agree with the editor's comment and changed the sentence accordingly.

**Comment 9**

l254: "physics" -> "physics parameterizations"

**Response:** Thank you for the comment.

We agree with the editor's comment and changed the expression as explained in 3.

**Comment 10**

l255: "Hence, the SST definition presented Section 2.3" -> "Hence we use the SST definition presented in Section 2.3"

**Response:** Thank you for the comment.

We agree with the editor's comment and changed the sentence accordingly.

**Comment 11**

l274: The sentence starting with "The proposed background state" is confusing, and I'm not sure how to interpret what you intended to communicate. Part of my confusion is that you start the sentence talking about the background state (which I take to be the initial conditions of the model state variables), but then the sentence discusses parameterization details: it seems like the sentence is implying that the background state and parameterizations are equivalent. This sentence should be revised.

**Response:** Thank you for the comment.

We agree with the editor's comment and changed the text as follow:

> The proposed solution has been implemented as a new idealised case (indicated by the NTESTCASE parameter in OpenIFS), where the model state variables are initialised based on the equations for geopotential, virtual temperature, the horizontal wind components and, in the case of moist simulations, the specific humidity that were derived above. Once the initial values of the state variables are defined in the model, the OpenIFS simulations are integrated forward in time on an aquaplanet.

**Comment 12**

l286: "humidity (HDIRQ) diffusions) and the other are set to zero." ->" humidity (HDIRQ) diffusions, and the other coefficients are set to zero.

**Response:** Thank you for the comment.

We agree with the editor's comment and changed the sentence accordingly.

**Comment 13**

l299: "A standalone version have been developed" -> "A standalone version has been developed"

**Response:** Thank you for the comment.

We agree with the editor's comment and changed the sentence accordingly.

**Comment 14**

l308: it looks like there is an extra ")" after 2023-12-05

**Response:** Thank you for the comment.

We agree with the editor's comment and changed the sentence accordingly.

**Comment 15**

l340: it has been hundreds of lines since BWS was defined, and it isn't a terribly common acronym; I had forgotten the acronym's meaning by the time I got to this sentence. It might be better to use the whole phrase here rather than the acronym.

**Response:** Thank you for the comment.

We agree with the editor's comment and changed the sentence accordingly.

> ### Comment 16
>
> l347: "to enable to first" -> "to enable the first"

**Response:** Thank you for the comment.

We agree with the editor's comment and changed the sentence accordingly.

> ### Comment 17
>
> Figure 4e: in the vicinity of latitudes -20 and 20, theta-E is non-monotonic. It decreases from approximately 345K at the surface to 240K at about 500 hPa, and then it increases again after that. This seems to suggest there is conditional instability in this region. Line 330 seems to indicate that you are aiming to create the theta-E profiles such that they increase monotonically everywhere. Does the language around line 330 need to be revised?

**Response:** Thank you for the comment.

We agree with the editor's comment and have revised section 4 as follow:

> For the initial state to be absolutely stable to dry and saturated vertical displacements (static stability), equivalent potential temperature must increase with height everywhere. In the situation where equivalent potential temperature decreases with height, conditional instability is present, meaning that the atmosphere is stable to displacements of dry and unsaturated air parcels but unstable to displacements of saturated air parcels. If potential temperature decreases with height, then the atmosphere is absolutely unstable - both dry and saturated displacements are

unstable. Thus, for the initial state to be absolutely stable potential temperature must increase with height and the Brunt-Väisälä frequency must be positive. Regions where equivalent potential temperature decreases with height are also stable and acceptable in the initial state as long as these regions are not saturated.

**Comment 18**

line 480: the sentence refers to 'friction' twice, which seems like it might be a typo

**Response:** Thank you for the comment.

We agree with the editor's comment and changed the sentence accordingly.

**Comment 19**

line 536: "However, as this CAPE can only be addressed" -> "However, as this CAPE can only be accessed"

**Response:** Thank you for the comment.

We agree with the editor's comment and changed the sentence as follow:

However, as this CAPE can only be released once the surface parcel reaches saturation or is substantially lifted, and the fact that this area is not affected by the baroclinic wave, this does not influence the meteorological stability of our setup.